JCB Journal of Cell Biology

# Centriole growth is limited by the Cdk/Cyclin-dependent phosphorylation of Ana2/STIL

Thomas L. Steinacker[1]*, Siu-Shing Wong[1]*, Zsofia A. Novak[1], Saroj Saurya[1], Lisa Gartenmann[1], Eline J.H. van Houtum[1], Judith R. Sayers[1], B. Christoffer Lagerholm[2], and Jordan W. Raff[1]

**Centrioles duplicate once per cell cycle, but it is unclear how daughter centrioles assemble at the right time and place and grow to the right size. Here, we show that in *Drosophila* embryos the cytoplasmic concentrations of the key centriole assembly proteins Asl, Plk4, Ana2, Sas-6, and Sas-4 are low, but remain constant throughout the assembly process—indicating that none of them are limiting for centriole assembly. The cytoplasmic diffusion rate of Ana2/STIL, however, increased significantly toward the end of S-phase as Cdk/Cyclin activity in the embryo increased. A mutant form of Ana2 that cannot be phosphorylated by Cdk/Cyclins did not exhibit this diffusion change and allowed daughter centrioles to grow for an extended period. Thus, the Cdk/Cyclin-dependent phosphorylation of Ana2 seems to reduce the efficiency of daughter centriole assembly toward the end of S-phase. This helps to ensure that daughter centrioles stop growing at the correct time, and presumably also helps to explain why centrioles cannot duplicate during mitosis.**

## Introduction

Centrioles form cilia and centrosomes, two organelles that are important organizers of the cell (Nigg and Raff, 2009; Conduit et al., 2015; Loncarek and Bettencourt-Dias, 2018; Breslow and Holland, 2019; Bornens, 2021; Vasquez-Limeta and Loncarek, 2021). Most newborn cells inherit a single pair of centrioles and, in cycling cells, these centrioles separate and then duplicate when a new daughter centriole grows from the side of each existing mother centriole. It is well established that centrioles normally duplicate in the S-phase, and that mitosis appears to be refractory for duplication (Lacey et al., 1999; Hinchcliffe et al., 1999; Hinchcliffe and Sluder, 2001). The mechanisms that enforce this strict cell cycle regulation remain largely obscure.

Recent studies have identified a conserved pathway of centriole duplication (Nigg and Holland, 2018; Gönczy and Hatzopoulos, 2019). Plk4 is the key enzyme that promotes centriole assembly (Bettencourt-Dias et al., 2005; Habedanck et al., 2005), and it is recruited to the centrioles by Asl in flies (Dzhindzhev et al., 2010), SPD-2 in worms (Kemp et al., 2004; Shimanovskaya et al., 2014), and a combination of the two (CEP152 and CEP192, respectively) in humans (Sonnen et al., 2013; Kim et al., 2013). Plk4 is initially recruited in a ring around the mother centriole, but it rapidly becomes focused at a single site on the mother centriole that specifies where the daughter centriole will assemble (Arquint and Nigg, 2016; Leda

et al., 2018; Takao et al., 2019; Yamamoto and Kitagawa, 2021). Plk4 recruits Ana2/STIL (fly/human) to centrioles, re-enforcing the specific localization of Plk4 and activating Plk4 to phosphorylate Ana2/STIL to further promote Ana2's recruitment and also its interaction with Sas-6 (Dzhindzhev et al., 2014, 2017; Ohta et al., 2014, 2018; Kratz et al., 2015; Moyer et al., 2015; Moyer and Holland, 2019). Sas-6 and Ana2 cooperate to initiate the formation of the central cartwheel, upon which the rest of the centriole is assembled (Kitagawa et al., 2011; van Breugel et al., 2011; Stevens et al., 2010b).

It is unclear how daughter centrioles grow to the correct size. In flies and worms, the central cartwheel and centriole MTs grow to approximately the same size (Schwarz et al., 2018; Gonzalez et al., 1998), and in the syncytial embryos of *Drosophila* both structures appear to abruptly stop growing in mid-late S-phase (Aydogan et al., 2018). The centriolar levels of Plk4 oscillate during each round of centriole duplication in fly embryos and human cultured cells (Aydogan et al., 2020; Takao et al., 2019). In fly embryos, this oscillation is normally entrained by the Cdk/Cyclin cell cycle oscillator (CCO) that times the rapid nuclear cycles in these syncytial embryos, and this ensures that Plk4 is maximally recruited to the centrioles in the late-mitosis/early S-phase when the daughter centrioles are starting to grow. However, the rather abrupt cessation of

......................................................................................................................................................................................................................

[1]Sir William Dunn School of Pathology, University of Oxford, Oxford, UK;   [2]Kennedy Institute of Rheumatology, University of Oxford, Oxford, UK.

*T.L. Steinacker and S.-S. Wong contributed equally to this paper.   Correspondence to Jordan W. Raff: jordan.raff@path.ox.ac.uk

T.L. Steinacker's present address is Institute of Molecular Biotechnology of the Austrian Academy of Sciences (IMBA), Vienna Biocenter (VBC), Vienna, Austria.   J.R. Sayers's present address is Department of Physiology, Anatomy and Genetics (DPAG), University of Oxford, Oxford, UK.

centriole growth in fly embryos during mid-late-S-phase (Aydogan et al., 2018) may be hard to reconcile with the more gradual decline in centriolar Plk4 levels (Aydogan et al., 2020). We suspected, therefore, that other mechanisms might work together with the Plk4 oscillation to ensure that the centrioles in fly embryos stop growing in late S-phase.

Quantitative mass spectroscopy has revealed that several key centriole assembly proteins (e.g., CEP152/Asl, PLK4/Plk4, SAS6/Sas-6, STIL/Ana2, and CPAP/Sas-4; human/fly nomenclature) are present at low levels in human cells (Bauer et al., 2016), raising the possibility that one or more of these proteins might become depleted from the cytoplasm as daughter centriole assembly proceeds, potentially contributing to the cessation of centriole growth. In worm embryos, such a "limiting component" mechanism is thought to set centrosome size, as the amount of pericentriolar material (PCM) that assembles around the centrioles appears to be set by a limiting pool of the key PCM-building block SPD-2 (Decker et al., 2011)—a protein that in worms is also essential for centriole duplication (Kemp et al., 2004; Pelletier et al., 2004). An alternative mechanism for limiting centriole growth has been suggested in human cells, where Cdk1/Cyclin B can inhibit centriole duplication by directly competing with Plk4 for binding to the central coiled-coil domain (CC) of STIL/Ana2 (Zitouni et al., 2016). In the early fly embryo, such a mechanism should lead to the inhibition of centriole growth as Cdk/Cyclin levels rise during the S-phase (Deneke et al., 2016). There is some question, however, as to whether the interaction between Plk4 and the STIL/Ana2 CC is essential as the CC is also required for STIL multimerization—which is essential for STIL/Ana2 function (Cottee et al., 2015; David et al., 2016)—and structural studies suggest that multimerization is incompatible with binding to PLK4 (Cottee et al., 2017). Moreover, Ana2/STIL proteins can also bind Plk4 through their C-terminal regions, independently of the CC (Ohta et al., 2018; McLamarrah et al., 2018). Thus, direct competition between Plk4 and Cdk1/Cyclin B for binding to the CC of STIL/Ana2 seems unlikely to be a universal mechanism that suppresses centriole duplication when Cdk1/Cyclin B levels are high.

Here, we have used fluorescence correlation spectroscopy (FCS; Kim et al., 2007) and peak counting spectroscopy (PeCoS; Aydogan et al., 2020) to monitor how the cytoplasmic concentration and/or biophysical characteristics of the core centriole duplication proteins in Drosophila (Asl, Plk4, Sas-6, Ana2, and Sas-4) change during the nuclear cycle in the living early embryos. We find that although the cytoplasmic concentration of all these proteins is low (likely ~1–30 nM range), their concentration remains constant as the centrioles assemble. This suggests that none of these proteins act as limiting components that slow centriole growth because they become depleted from the cytoplasm. Strikingly, however, we noticed that the cytoplasmic diffusion rate of Ana2 increased significantly toward the end of the S-phase, and this seemed to depend, at least in part, upon its phosphorylation by Cdk/Cyclins. This phosphorylation appears to inhibit Ana2's ability to promote centriole duplication in the late S-phase when Cdk/Cyclin levels are rising rapidly for the preparation of mitosis. We propose that this novel mechanism

helps to ensure that centrioles stop growing at the appropriate time, and likely also helps to ensure that centrioles cannot duplicate in mitosis when Cdk/Cyclin activity is maximal.

## Results

### Generating tools for FCS measurements

To analyze the behavior of the core duplication proteins under conditions as close to physiological as possible, we fluorescently tagged Asl, Plk4, Sas-6, Ana2, and Sas-4 at their endogenous loci with monomeric NeonGreen (mNG) using CRISPR/Cas9 (Port et al., 2014). The fusion proteins all localized to centrioles (Fig. 1 A), and Western blotting confirmed that they were expressed at similar levels to their endogenous proteins, except for mNG-Ana2 and Ana2-mNG, which appeared to be overexpressed by ~2–4X when compared to the endogenous untagged protein (Fig. 1 B); note that we could not examine Plk4 in this way as it could not be detected by Western blotting. We failed to generate a Plk4-mNG knock-in, and an mNG-Plk4 knock-in line laid embryos that exhibited consistent centriole overduplication, suggesting that the fusion was overexpressed (yellow arrows, Fig. 1 A ii). We, therefore, chose to further analyze Plk4 behavior using a transgenic line (ePlk4-mNG; Aydogan et al., 2020) in which the centrioles do not overduplicate (red box, Fig. 1 A ii). We also examined transgenic lines expressing either monomeric NeonGreen (mNG) or dimeric NeonGreen (dNG) expressed from the Sas-6 promoter that were not fused to any target protein; these proteins did not detectably localize to centrioles, and they serve as inert controls that should not interact physiologically with other proteins in the cytoplasm.

### The cytoplasmic concentration of the Drosophila core centriole duplication proteins is low but remains relatively constant as daughter centrioles assemble in S-phase

In worm embryos, a limiting pool of SPD-2 is thought to set centrosome size as it becomes sequestered at the growing centrosomes and so depleted from the cytoplasm (Decker et al., 2011). To test whether a similar mechanism might help to limit centriole growth in the early Drosophila embryo, we used FCS to monitor the cytoplasmic concentration of the core duplication proteins as the centrioles assembled during the S-phase of nuclear cycle 12. Control experiments in which we altered the genetic dosage of fluorescent fusion proteins confirmed that FCS can be used to measure cytoplasmic concentration changes in the early Drosophila embryo (Fig. S1, A and B).

As a control, we first examined the behavior of untagged mNG and dNG expressed from the Sas-6 promoter (Fig. 2 A). In both cases, the concentration of mNG or dNG remained relatively constant throughout nuclear cycle 12, although there was a tendency for their concentration to dip slightly during the early S-phase, to rise slightly as the embryos entered mitosis, and then dip again as the embryos entered the next cycle. These proteins are biologically inert, so we suspect that these minor fluctuations occur because the biophysical properties of the cytoplasm change slightly as the embryos progress through the nuclear cycle. In support of this possibility, we observed a broadly similar pattern when we examined the concentration of

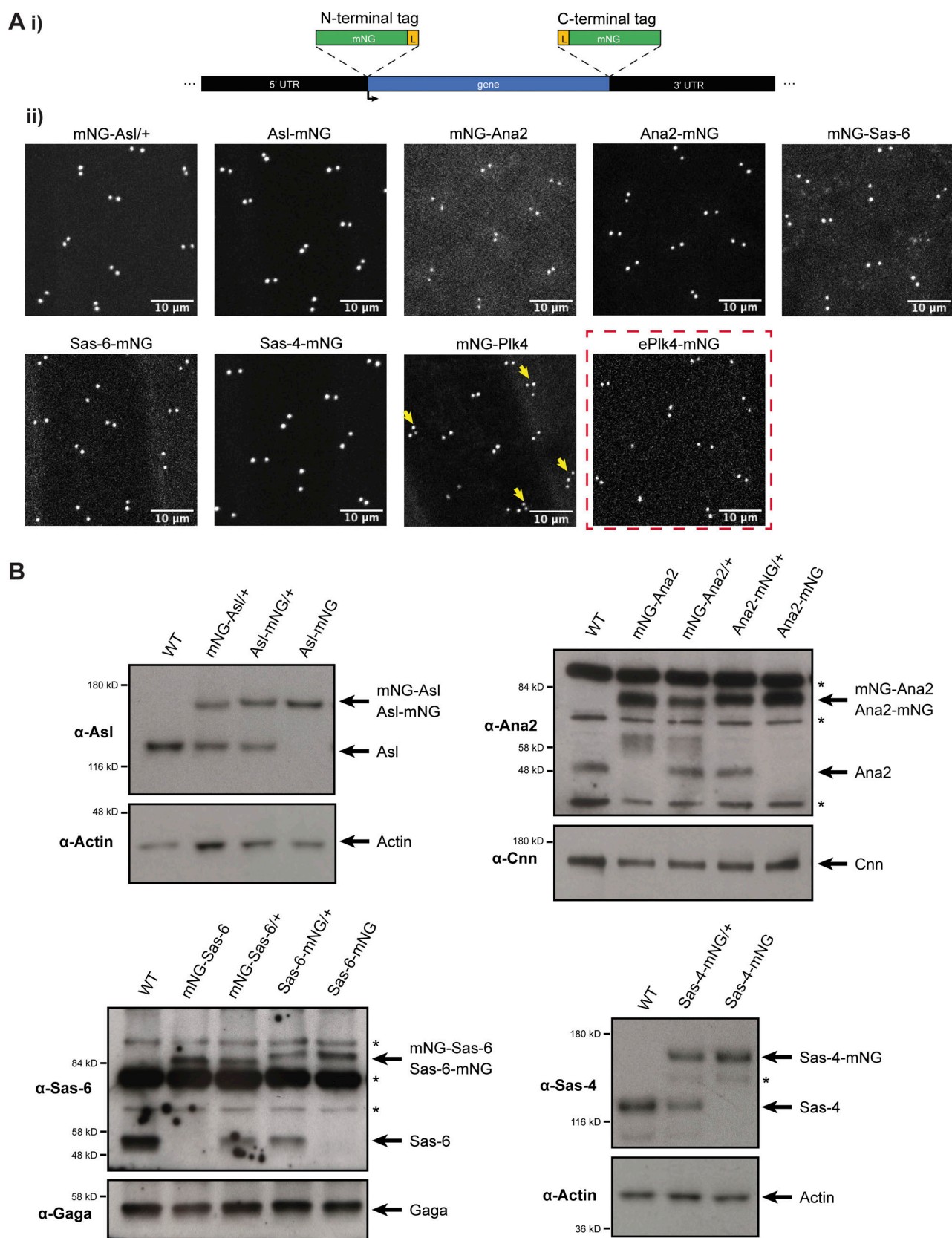

Figure 1. **Generation of endogenously mNG-tagged centriolar proteins. (A i)** Schematic illustration of the strategy to "knock-in" mNG at the N- or C-terminus of an endogenous locus; (L) is a short linker sequence. **(ii)** Images show the centriolar localization of the mNG-tagged CRISPR/Cas9-mediated knock-ins in living syncytial embryos (all images acquired in the early S-phase of nuclear cycle 12). N-terminally tagged mNG-Asl was not viable so it was expressed in a heterozygous (*mNG-Asl/+*) background. N-terminally tagged mNG-Plk4 consistently caused centriole overduplication (yellow arrows), so in

subsequent experiments, we used a P-element insertion line of Plk4-mNG expressed from its endogenous promoter in the *Plk4*[−/−] mutant background, (ePlk4-mNG, red dashed box). **(B)** Western blots show the expression levels of CRISPR/Cas9 knock-in lines and their cognate untagged endogenous proteins in 0–2 h old embryos. Prominent non-specific bands are highlighted (*); Actin, Cnn, and the Gaga transcription factor are shown as loading controls. A representative blot is shown from at least two technical repeats.

Asl-mNG, mNG-Sas-6, Sas-6-mNG, Sas-4-mNG, mNG-Ana2, or Ana2-mNG (Fig. 2 B). The average cytoplasmic concentration of all these proteins was low: ~7–15 nM for the Asl, Sas-6, and Sas-4 knock-in lines, and 18–42 nM for the Ana2 knock-in lines. As the mNG-knock-ins at the Ana2 locus appear to be ~2–4X overexpressed, we conclude that these core centriole duplication proteins are likely present in the ~5–20 nM concentration range, which seems surprisingly low, but is in agreement with previous estimates from human cells (Bauer et al., 2016; see Discussion). Importantly, the concentration of all these proteins remained relatively constant over the entire nuclear cycle, while exhibiting the same general tendency as the mNG and dNG controls to fluctuate slightly.

We showed previously that the concentration of a Plk4-mNG fusion driven transgenically from its own promoter (ePlk4-mNG) was too low to be measured by FCS (Aydogan et al., 2020), and this was also true of our mNG-Plk4 CRISPR/Cas9 knock-in line, even though this protein appeared to be overexpressed, leading to centriole overduplication (Fig. 1 A). Interestingly, a previous study using a similar microscopy setup but a different mNG-Plk4 knock-in line used FCS to estimate a concentration of ~7–8 nM (Nabais et al., 2021). As we could not use FCS, we used PeCoS (Aydogan et al., 2020) to monitor how the cytoplasmic concentration of the transgenically expressed ePlk4-mNG fusion protein varied during nuclear cycle 12 (as the centrioles do not overduplicate in this line). This data was noisy, but we detected no significant change in the cytoplasmic concentration of ePlk4-mNG during nuclear cycle 12 (Fig. 2 C). Taken together, these data suggest that the cytoplasmic concentration of all the core centriole duplication proteins remains relatively constant during nuclear cycle 12, meaning that none of them are likely to act as limiting components for centriole growth.

### Sas-6 appears to be monomeric in the cytoplasm, while Ana2 appears to be multimeric

Structural studies strongly suggest that Sas-6 forms a dimer that is the key structural building block of the cartwheel (Kitagawa et al., 2011; van Breugel et al., 2011). The ability of Ana2/STIL proteins to multimerize also appears to be essential for their function (Arquint et al., 2015; Cottee et al., 2015; Rogala et al., 2015; David et al., 2016), with the recombinant central coiled-coiled region of *Drosophila* Ana2 and *C. elegans* SAS-5 (the worm homolog of Ana2/STIL) forming either a tetramer (Cottee et al., 2015) or a trimer (Rogala et al., 2015), respectively in vitro. These in vitro studies, however, were usually performed at protein concentrations in the 10–1,000 μM range, whereas our FCS studies suggest that these proteins are present in the embryo in the ~10–20 nM range. We, therefore, used FCS to monitor Sas-6 and Ana2's "photon-count rate per molecule" (CPM). This is the average number of photons generated by each fluorescently tagged molecule that passes through the FCS

observation volume, so the CPM of a fluorescent dimer should be nearly twice that of a fluorescent monomer (the photochemistry means the fluorescence will not precisely double; Dunsing et al., 2018). As a control, dNG exhibited a CPM that was ~1.7-fold higher than mNG (Fig. S2 A). Interestingly, Ana2-mNG and mNG-Ana2 had a CPM that was similar to dNG (Fig. S2 A), suggesting that they exist in the cytoplasm as homo-oligomers that, on average, are dimers. In contrast, Sas-6-mNG and mNG-Sas-6 had a CPM that was approximately equal to mNG, suggesting that, surprisingly, the Sas-6-fusions are predominantly monomeric in the cytoplasm (Fig. S2 A; see Discussion). Asl-mNG and Sas-4-mNG also exhibited a CPM that was most similar to the monomeric control, and the CPM of all the proteins we examined did not change significantly throughout nuclear cycle 12 (Fig. S2, B and C).

### The cytoplasmic diffusion rate of Ana2 increases as embryos exit S-phase and enter mitosis

To test whether any of the core duplication proteins might change their biophysical properties as the centrioles assembled, we used FCS to see if their diffusion rates changed during nuclear cycle 12 (Fig. 3; note that PeCoS does not allow us to extract this information for Plk4). The diffusion rate of the inert mNG and dNG controls did not change significantly over the cycle (Fig. 3 A), but for Asl, Sas-6, and Sas-4, it tended to increase slightly as S-phase progressed, and then decrease slightly during mitosis and into the next nuclear cycle (Fig. 3 B i–iv). This tendency was not, or was only just, statistically significant, but it was consistent, and no similar trend was observed with the mNG and dNG controls. This suggests that the average cytoplasmic diffusion rate of these three core duplication proteins may increase subtly as S-phase progresses—perhaps indicating that, on average, their ability to interact with other cytoplasmic proteins gradually decreases during the assembly process.

Strikingly, and in contrast to the other core duplication proteins, the diffusion rate of both the mNG-Ana2 and Ana2-mNG fusions remained relatively constant in the early-mid S-phase, but then increased sharply in the late S-phase as the embryos prepared to enter mitosis, before falling sharply again at the start of the next cycle (pink boxes, Fig. 3 B v and vi).

### The change in Ana2 diffusion rate during the nuclear cycle does not appear to depend on Ana2's ability to bind to Sas-6

We wanted to test if Ana2's ability to multimerize or to interact with Sas-6 was required for the change in Ana2's diffusion rate during the nuclear cycle. The central coiled-coil (CC) domain of Ana2/STIL proteins is essential for their homo-oligomerization (Arquint et al., 2015; Cottee et al., 2015; Rogala et al., 2015), while the conserved STAN domain is required for Ana2's interaction with Sas-6 (Dzhindzhev et al., 2014; Ohta et al., 2014; Kratz et al., 2015; Moyer et al., 2015). We generated flies transgenically

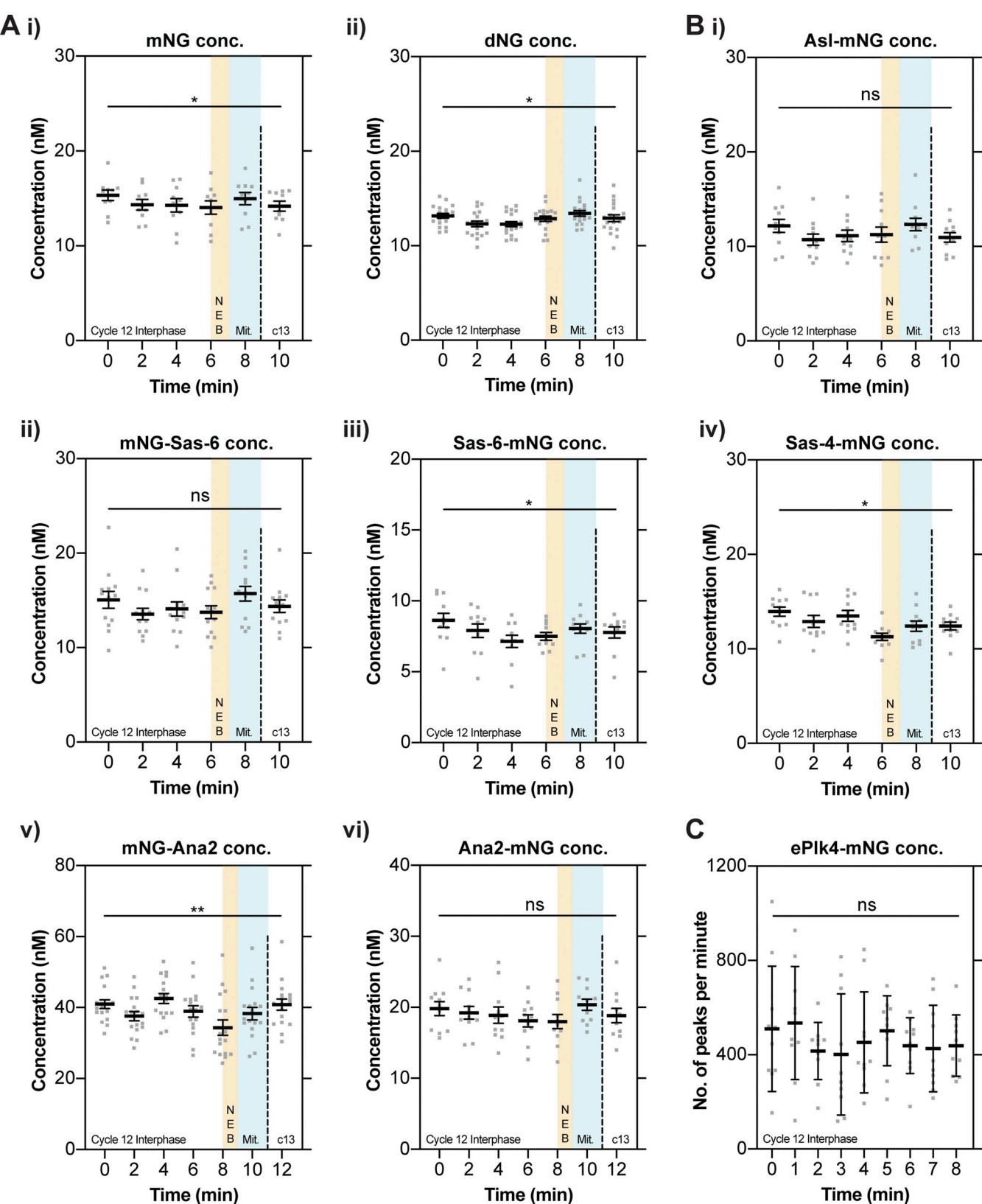

Figure 2. **The cytoplasmic concentration of the core centriole duplication proteins does not change dramatically as daughter centrioles assemble during nuclear cycle 12. (A and B)** Graphs show cytoplasmic FCS concentration measurements (mean ± SEM) of either mNG or dNG controls (A) or mNG-fusions to the core centriole duplication proteins (B). Measurements were taken every 2 min from the start of nuclear cycle 12. The timing window of NEB is depicted in yellow and mitosis in green. Each data point represents the average of 4–6× 10-s recordings from an individual embryo ($N \geq 10$). **(C)** The graph shows ePlk4-mNG PeCoS measurements (mean ± SD) taken at 60-s intervals from the start of nuclear cycle 12. Each data point represents an individual 60 s PeCoS measurement ($N = 10$). Statistical significance was assessed using a paired one-way ANOVA test (for Gaussian-distributed data) or a Friedman test (**, $P < 0.01$; *, $P < 0.05$).

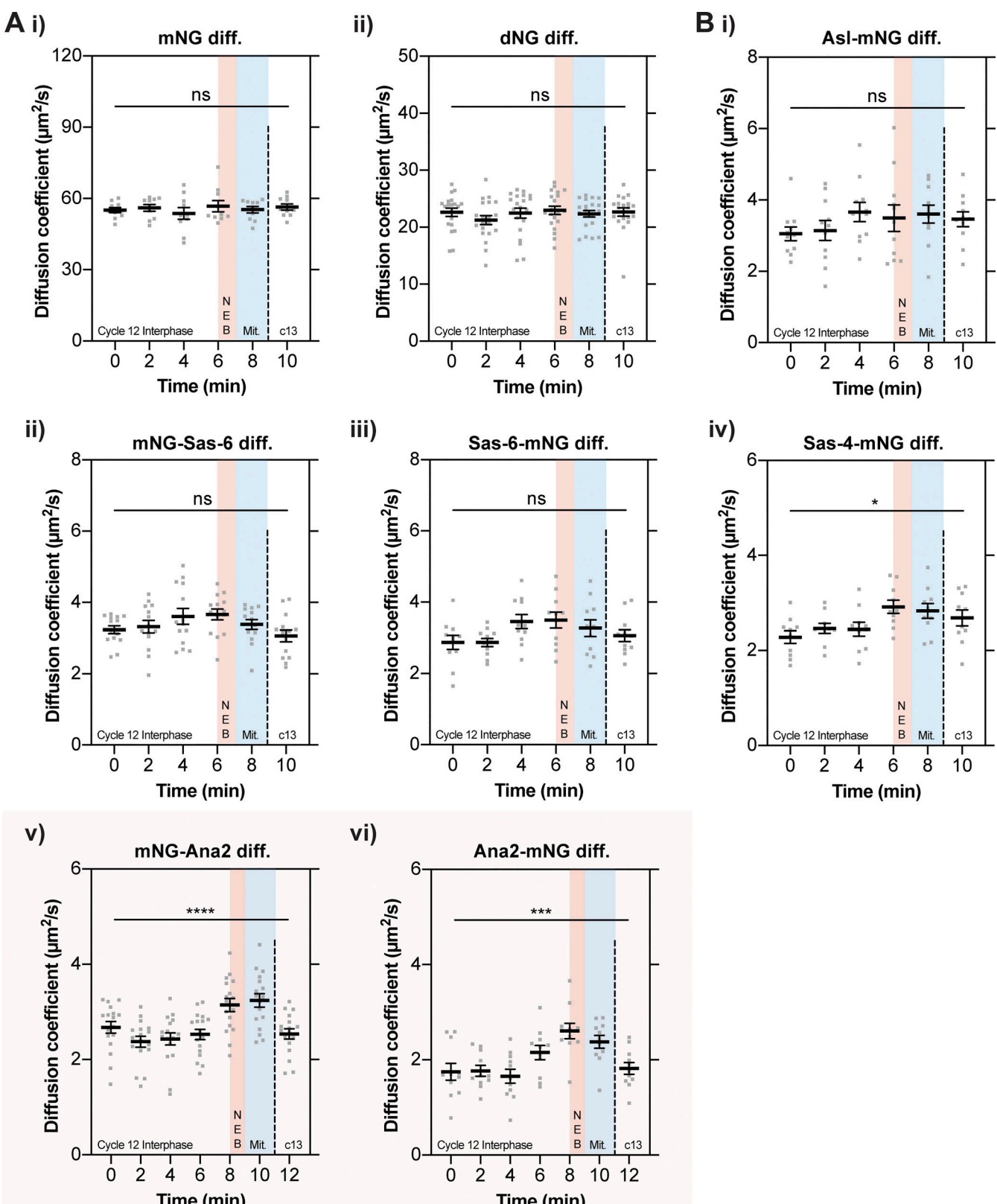

**Figure 3.** **The cytoplasmic diffusion rate of Ana2 changes significantly as embryos exit S-phase. (A and B)** Graphs show cytoplasmic FCS diffusion rate measurements (mean ± SEM) of either mNG or dNG controls (A) or mNG-fusions to the core centriole duplication proteins (B). Measurements were taken every 2 min from the start of nuclear cycle 12. The timing window of NEB is depicted in red, and of mitosis in blue. Each data point represents the average of 4–6× 10-s recordings from an individual embryo ($N ≥ 10$). The mNG-Ana2 and Ana2-mNG graphs are boxed in pink, as these proteins showed the most dramatic change in diffusion rates during the cycle. Statistical significance was assessed using a paired one-way ANOVA test (for Gaussian-distributed data) or a Friedman test (****, $P < 0.0001$; ***, $P < 0.001$; *, $P < 0.05$).

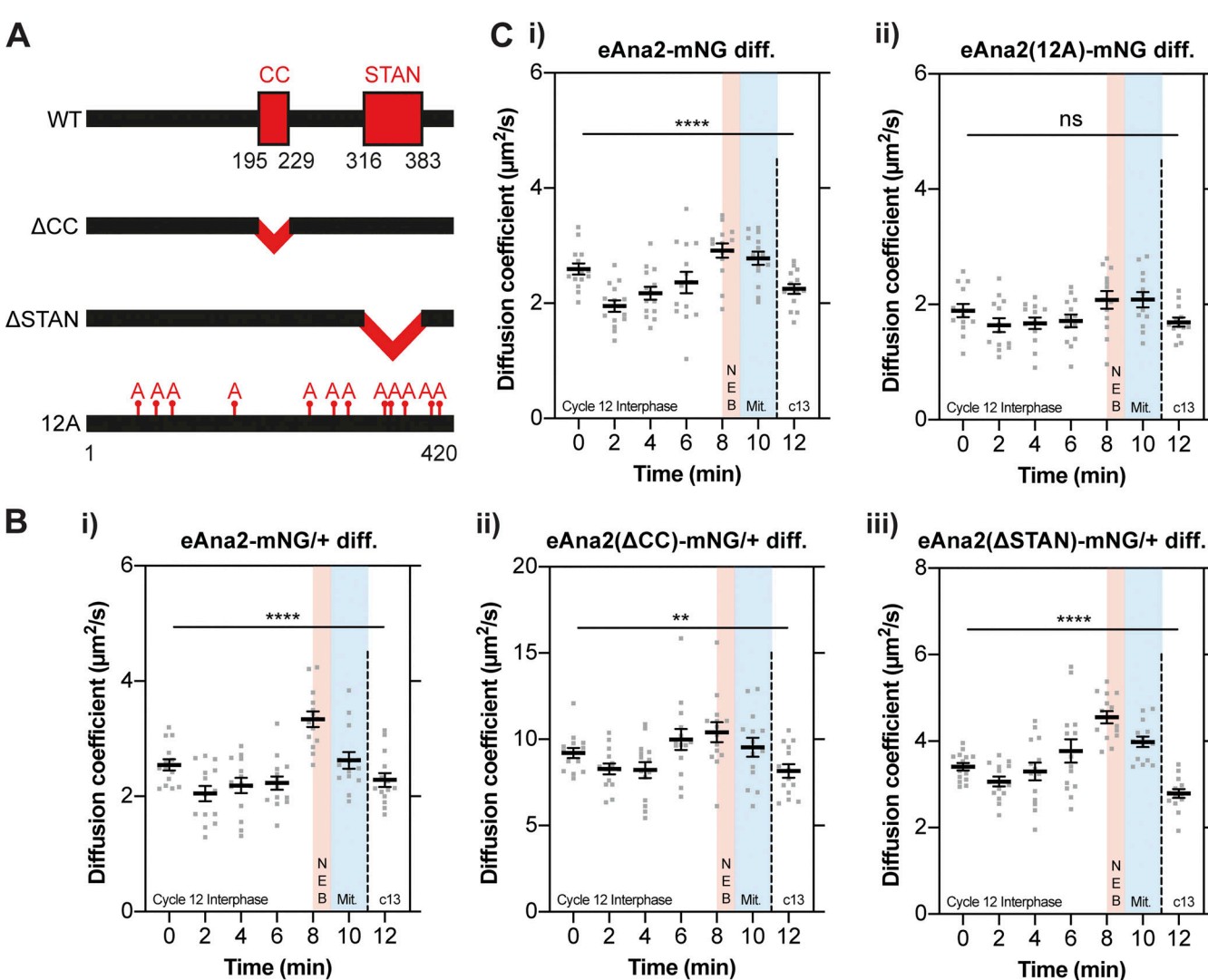

Figure 4. **Ana2's change in diffusion rate does not appear to depend on the CC or STAN domain, but this change is perturbed if Ana2 cannot be phosphorylated by Cdk/Cyclins. (A)** Schematic illustration of the Ana2 protein and the deletion/mutant forms analyzed in this study: central coiled-coil (CC) domain (aa195-229); STil/ANa2 (STAN) domain (aa316-383); the 12 S/T residues in S/T-P motifs that were mutated to Alanine. **(B and C)** Graphs show cytoplasmic FCS diffusion measurements (mean ± SEM) in embryos laid by females of the following genotypes: B (i) eAna2-mNG/+; B (ii) eAna2(ΔCC)-mNG/+; B (iii) eAna2(ΔSTAN)-mNG; C (i) eAna2-mNG; C (ii) eAna2(12A)-mNG. Measurements were taken every 2 min from the start of nuclear cycle 12. The timing window of NEB is depicted in red and mitosis in blue. Each data point represents the average of 4–6× 10-s recordings from an individual embryo ($N \geq 13$). Statistical significance was assessed using a paired one-way ANOVA test (for Gaussian-distributed data) or a Friedman test (****, $P < 0.0001$; **, $P < 0.01$).

expressing forms of Ana2 in which either the CC (eAna2(ΔCC)-mNG) or STAN domain (eAna2(ΔSTAN)-mNG) was deleted (Fig. 4 A).

As Ana2 is essential for centriole duplication, $ana2^{-/-}$ mutant females lack centrioles and are uncoordinated due to the lack of cilia in their sensory neurons—so they cannot mate or lay embryos (Stevens et al., 2010a; Basto et al., 2006). As the CC and STAN domains are essential for Ana2 function (Cottee et al., 2015), the mutant transgenes did not rescue this uncoordinated phenotype. To analyze the behavior of these proteins in embryos we, therefore, had to generate heterozygous fly lines expressing one copy of the mNG-tagged WT or mutant Ana2 together with one copy of the WT untagged *ana2* gene. All the fusion proteins were expressed at broadly similar levels to each other and the untagged endogenous protein in embryos,

although Ana2(ΔCC)-mNG appeared to be slightly destabilized and Ana2(ΔSTAN)-mNG slightly stabilized (Fig. S1 C). The average diffusion rate of both truncated proteins at the start of S-phase was elevated compared to WT eAna2-mNG—from ~2.5 µm²/s (WT) to ~9 µm²/s (ΔCC) and ~3.5 µm²/s (ΔSTAN)—but the significant increase in diffusion rate in late-S-phase/early-mitosis was still detectable, although this was somewhat suppressed for the ΔCC mutant (Fig. 4 B). Thus, the change in Ana2's cytoplasmic diffusion rate may be enhanced if the protein can homo-oligomerize, but it does not appear to depend on its interaction with Sas-6. An important caveat to these experiments is that they are performed in the presence of WT protein, which may oligomerize with the mutant proteins. The rapid diffusion of Ana2-ΔCC suggests that this protein at least does not form homo-oligomers, consistent with previous structural studies

(Cottee et al., 2015). Nevertheless, we remain cautious in drawing definitive conclusions from these experiments.

**The change in Ana2 diffusion during the nuclear cycle appears to depend, at least in part, on phosphorylation by Cdk/Cyclins**
We wondered whether the diffusion rate increase of Ana2 might depend upon its cell-cycle specific phosphorylation. CDK1-Cyclin B is a potential candidate kinase, as it can phosphorylate vertebrate STIL (Zitouni et al., 2016) and, in the early *Drosophila* embryo, Cdk/Cyclin activity gradually increases as S-phase progresses (Deneke et al., 2016). To test the potential role of Cdk1/Cyclin-dependent phosphorylation, we generated fly lines transgenically expressing a mutant form of Ana2 in which all 12 potential Cdk phosphorylation sites (S/T-P motifs) were mutated to non-phosphorylatable alanine (A-P; eAna2(12A)-mNG; Fig. 4 A and Fig. S3). We think it unlikely that Cdk/Cyclins normally phosphorylate all 12 of these sites to regulate Ana2 function but, given that we do not know the potentially relevant sites, this approach allows us to test the function of a form of Ana2 that cannot be phosphorylated by Cdk/Cyclins. Importantly, mass spectroscopy studies have identified peptides phosphorylated at 10 of these 12 sites in extracts from S2 cells or embryos (McLamarrah et al., 2018; Dzhindzhev et al., 2017; Fig. S3 A), indicating that Cdk/Cyclins could potentially phosphorylate Ana2 in vivo. Moreover, short peptides containing two well-conserved sites (S284 and T301; Fig. S3 A) can be specifically and efficiently phosphorylated by recombinant Cdk1/Cyclin B in vitro (Fig. S4).

The eAna2(12A)-mNG transgene fully rescued the defects in $ana2^{-/-}$ flies caused by the lack of centrioles: rescued flies were as coordinated as WT controls and laid embryos that hatched at similar rates (Fig. S5, A and B). Moreover, we detected no centriole defects in EM studies of $ana2^{-/-}$ mutant wing disc cells rescued by transgenically expressing an untagged version of eAna2(12A) (Fig. S5 C). We conclude that the Ana2(12A) protein is largely functional, and that centriole duplication is not dramatically perturbed in fly cells when Ana2 cannot be phosphorylated by Cdk/Cyclins.

To test whether the behavior of Ana2(12A) might nevertheless be subtly altered, we used FCS to compare the cytoplasmic diffusion behavior of WT eAna2-mNG and eAna2(12A)-mNG throughout the nuclear cycle 12. Transgenic WT eAna2-mNG was expressed at similar levels to the Ana2-mNG CRISPR knock-in line (Fig. S1 D), and it exhibited the same dramatic rise and fall in diffusion rate (Fig. 4, B i and C i). The transgenic eAna2(12A)-mNG protein was expressed at similar levels (Fig. S5 D), but the rise and fall in its diffusion rate during nuclear cycle 12 was much less obvious and was not statistically significant (Fig. 4 C ii). We conclude that phosphorylation of Ana2 by Cdk/Cyclins could play a part in Ana2's cell cycle-specific diffusion change.

**Ana2(12A) accumulates at centrioles for an abnormally long period**
To test whether the 12A mutations influence Ana2's interaction with centrioles, we compared the dynamics of Ana2-mNG and eAna2(12A)-mNG centriolar recruitment during nuclear cycle 12

(Fig. 5 A). Similar to the other core centriole cartwheel protein Sas-6 (Aydogan et al., 2018), WT Ana2-mNG initially accumulated at centrioles in a near-linear fashion during early S-phase, but whereas eSas-6-GFP incorporation usually plateaued by ~ mid-S-phase (Aydogan et al., 2018), Ana2 continued to accumulate at the centrioles until ~1–2 min before NEB, when its levels peaked and then started to decline rapidly (black line, Fig. 5 A). There was a strong correlation (r > 0.98; P < 0.0001) between the period of Ana2 accumulation at the centriole and S-phase length over nuclear cycles 11–13 (Fig. 5 C). This suggests that the core Cdk/Cyclin cell cycle oscillator (CCO)—that drives the nuclear cycles in these embryos and sets S-phase length (Farrell and O'Farrell, 2014)—influences the timing of Ana2 recruitment to the centrioles, supporting our hypothesis that Ana2 could be a direct target of Cdk/Cyclins.

Surprisingly, eAna2(12A)-mNG was present at higher levels on centrioles than WT Ana2-mNG (red line, Fig. 5 A), even though eAna2(12A)-mNG was expressed at similar, or if anything slightly lower, levels than WT Ana2-mNG (Fig. S5 D). Moreover, although centriolar levels of WT Ana2-mNG peaked well before NEB, eAna2(12A)-mNG levels kept increasing until approximately the onset of mitosis (Fig. 5, A and B). This behavior is consistent with the possibility that Cdk1 normally phosphorylates Ana2 toward the end of S-phase to inhibit Ana2's recruitment to centrioles. Importantly, centriolar Ana2(12A)-mNG levels still started to decline once the embryos had actually entered mitosis (Fig. 5 A), so there was still a strong correlation (r > 0.91, P < 0.0001) between the period of Ana2(12A) growth and S-phase length (Fig. 5 C). This indicates that other mechanisms must normally help to ensure that Ana2 does not accumulate at centrioles during mitosis (e.g., perhaps the receptors that normally recruit Ana2 to centrioles also become phosphorylated during mitosis to inhibit Ana2 recruitment). These "other" mechanisms presumably explain why Ana2(12A) is still not recruited to centrioles efficiently during mitosis, and why centriole duplication appears largely unperturbed in embryos and cells expressing Ana2(12A)—even though the kinetics of Ana2(12A) recruitment are not normal.

**Centrioles grow for a longer period, but at a slower rate, in eAna2(12A) embryos**
To assess how Ana2(12A) might influence the assembly of the centriole cartwheel we analyzed the incorporation of the core centriole cartwheel protein Sas-6-mNG in embryos laid by females transgenically expressing two copies of untagged eAna2(12A) in the $ana2^{-/-}$ mutant background (Fig. 6). In WT embryos, we observed a similar Sas-6-mNG incorporation profile as we previously described for eSas-6-GFP (Aydogan et al., 2018), and regression analysis confirmed that this was best fit by a linear increase during early-mid-S-phase followed by a plateau (presumably when the daughter centrioles reach their final size; Fig. 6 A). Sas-6-mNG growth kinetics were significantly altered in embryos expressing Ana2(12A) (Fig. 6 A). Strikingly, the centrioles continued to incorporate Sas-6 for a significantly longer period (Fig. 6 B i and ii), consistent with our hypothesis that if Ana2 cannot be phosphorylated by Cdk1/Cyclin, its ability to promote centriole growth is not inhibited efficiently in late S-phase.

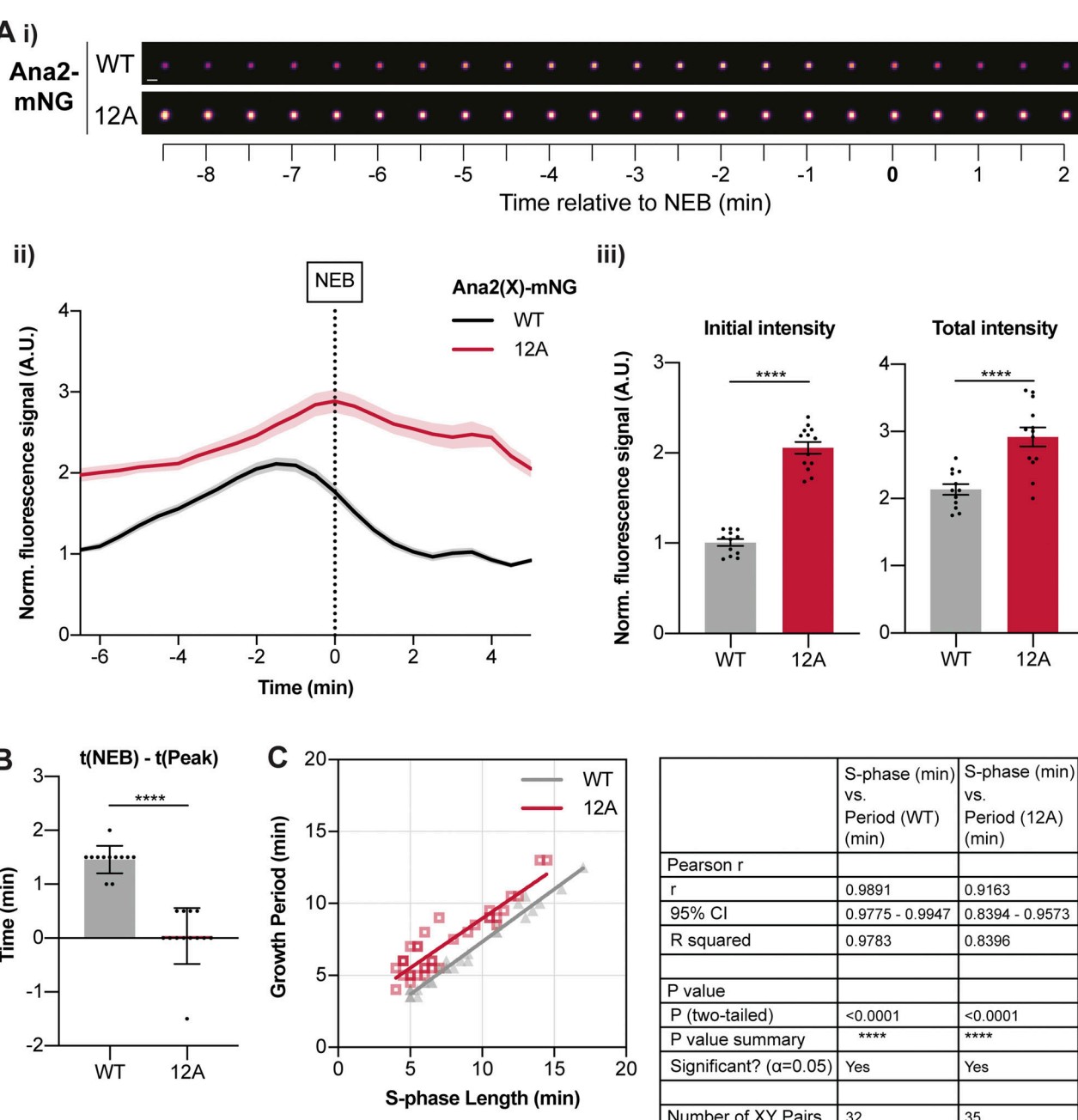

**Figure 5. eAna2(12A)-mNG exhibits an abnormal pattern of centriolar recruitment. (A i)** Images show the typical centriolar recruitment dynamics of WT Ana2-mNG or eAna2(12A)-mNG in an embryo during nuclear cycle 12—aligned to nuclear envelope breakdown (NEB; *t* = 0). Images were obtained by superimposing all the centrioles at each time point and averaging their fluorescence (scale bar = 1 μm). **(ii)** Graph shows the normalized (mean ± SEM) centriolar fluorescence levels of WT Ana2-mNG (black) and eAna2(12A)-mNG (red) during nuclear cycle 12 aligned to nuclear envelope breakdown (NEB; *t* = 0). *N* > 12 embryos; *n* ~ 100–150 centriole pairs per embryo. **(iii)** Bar charts quantify the normalised initial and maximal centriolar intensity (mean ± SEM). Each data point represents the average value of all centrioles measured in an individual embryo. **(B)** Quantification of the time (mean ± SD) at which Ana2 levels start to decrease at the centriole relative to NEB/mitosis. Statistical significance was assessed using an unpaired *t* test with Welch's correction (for Gaussian-distributed data) or a Mann-Whitney test (\*\*\*\*, P < 0.0001). **(C)** Scatterplot shows the correlation (obtained by linear regression of the data) between Ana2's growth period and S-phase length during nuclear cycles 11–13. *N* ≥ 10 embryos for each cycle, *n* ~ 70–90 (c11), *n* ~ 100–150 (c12), and *n* ~ 200–300 (c13) centriole pairs per embryo. Correlation strength was assessed using the Pearson's correlation coefficient.

Unexpectedly, however, significantly less Sas-6-mNG was recruited to centrioles in embryos expressing Ana2(12A) (Fig. 6, A and B iv–vi), and this was not due to any change in the total levels of Sas-6-mNG in the Ana2(12A) embryos (Fig. 6 C). Moreover, and potentially as a result of the decreased Sas-6

recruitment, the centrioles grew at a significantly slower rate in the presence of Ana2(12A) (Fig. 6, A and B iii). This finding is consistent with our previous observations that daughter centriole growth appears to be homeostatic: the centriole growth rate and growth period are inversely correlated so that if

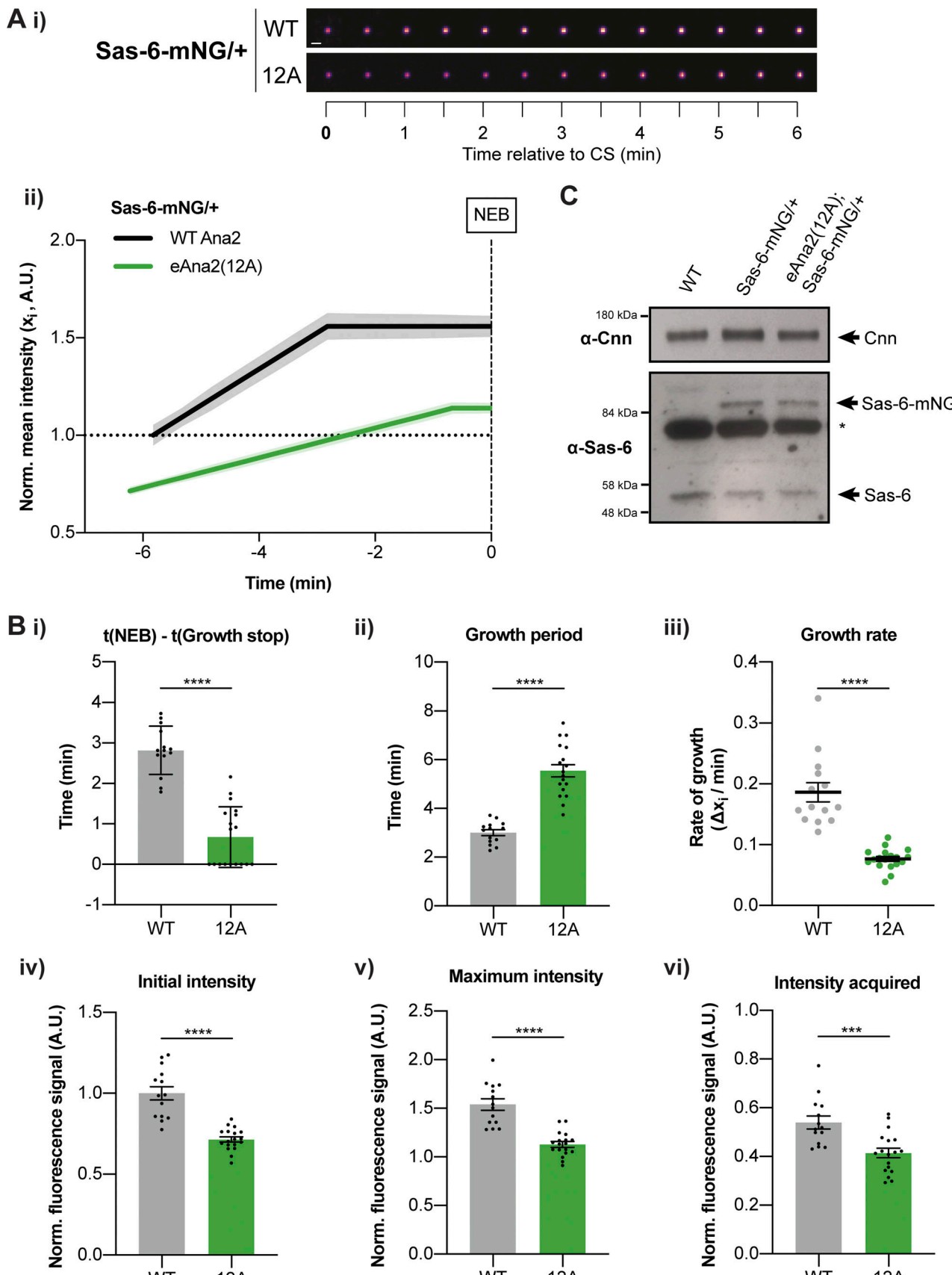

Figure 6. **Centrioles grow more slowly, but for a longer period, in the presence of eAna2(12A). (A i)** Images show the typical centriolar recruitment dynamics of Sas-6-mNG in a WT embryo or an embryo expressing eAna2(12A) during nuclear cycle 12—aligned to centriole separation at the start of S-phase

(CS; $t = 0$). Images were obtained by superimposing all the centrioles at each time point and averaging their fluorescence (scale bar = 1 µm). **(ii)** Graph shows the normalized (mean ± SEM) Sas-6-mNG centriole recruitment dynamics during nuclear cycle 12 in the presence of WT Ana2 (black) and eAna2(12A) (green) aligned to nuclear envelope breakdown (NEB; $t = 0$). N > 14 embryos, $n \sim 100$–150 centriole pairs per embryo. **(B)** Bar charts quantify and compare several centriole growth parameters (mean ± SEM) extracted from the data shown in (A ii). The values were derived from the fitted regression curve of the mean Sas-6-mNG intensity of each individual embryo. Each datapoint represents the average value of all the centriole pairs measured in each embryo. Statistical significance was assessed using an unpaired $t$ test with Welch's correction (****, $P < 0.0001$; ***, $P < 0.001$). **(C)** Western blot shows Sas-6 levels in WT embryos and embryos expressing one copy of Sas-6-mNG in either a WT or eAna2(12A) background. A prominent non-specific band is highlighted (*); Cnn is shown as loading control. A representative blot is shown from two technical repeats.

centrioles grow slowly, they tend to grow for a longer period and vice versa—so helping to ensure that centrioles grow to a consistent size (Aydogan et al., 2018). We currently do not understand why the expression of Ana2(12A) inhibits the recruitment of Sas-6 to centrioles (see Discussion), but it is fascinating that the expression of this mutant protein seems to induce a homeostatic response, with centrioles growing for a longer period, but at a slower rate. In embryos, this homeostasis is not perfect, and the centrioles appear to be slightly shorter in the presence of Ana2(12A); in somatic cells, where S-phase is much longer (presumably providing more time for adaptation), the centrioles grow to their normal size in the presence of Ana2(12A) (Fig. S5 C).

### Ana2(12A) does not appear to influence the behavior of the Plk4 oscillation at centrioles

We have previously shown that centriole growth kinetics are influenced by an oscillation in Plk4 levels at the centriole (Aydogan et al., 2020) and that, as in the Ana2(12A) embryos, the centrioles grow slowly but for a longer period when the genetic dose of Plk4 is halved. We, therefore, tested whether the centriolar Plk4 oscillation was altered in the Ana2(12A) embryos. Unfortunately, embryos laid by females expressing ePlk4-mNG and eAna2(12A) in the absence of any endogenous WT Ana2 failed to develop, so we had to perform this experiment in embryos laid by heterozygous females expressing one copy of eAna2(12A) in the presence of one copy of the endogenous WT *ana2* gene. The centriolar Plk4 oscillation in both sets of embryos was very similar, indicating that the expression of eAna2(12A) does not dramatically influence the Plk4 oscillation, at least under these conditions (Fig. 7).

### Ana2(12D/E) is not recruited efficiently to centrioles

Finally, we tested whether mutating the 12 S/T-P motifs in Ana2 to potentially phospho-mimicking D/E-P motifs influenced Ana2's behavior. The transgenic eAna2(12D/E)-mNG fusion was expressed at similar levels to WT Ana2-mNG and eAna2(12A)-mNG (Fig. S5 D), and it rescued the uncoordinated phenotype of *ana2⁻/⁻* mutant flies, indicating that, like Ana2(12A), Ana2(12D/E) can support centriole duplication and cilia assembly (Fig. S5 A). Unlike Ana2(12A), however, mutant females "rescued" by eAna2(12D/E)-mNG were sterile and laid embryos that failed to develop (Fig. S5 B). We have observed a similar phenotype previously with mutations in centriole duplication genes that inhibit the efficiency of centriole or centrosome assembly, but do not entirely prevent it (Cottee et al., 2015; Novak et al., 2016; Alvarez Rodrigo et al., 2019; Alvarez-Rodrigo et al., 2021). This seems to be because reducing the efficiency of centriole or

centrosome assembly is lethal to the early embryo (where centrioles and centrosomes have to assemble in only a few minutes), but not to somatic cells (where centrioles and centrosomes can assemble over a much longer period—presumably allowing these cells to better compensate for any inefficiency in the assembly process).

As embryos laid by females expressing only eAna2(12D/E)-mNG fail to develop, we examined this protein's centriole recruitment kinetics in embryos laid by females also expressing one copy of the endogenous untagged WT *ana2* gene. These embryos developed normally, but eAna2(12D/E)-mNG was recruited to centrioles very poorly (Fig. 8). This is consistent with our hypothesis that phosphorylation at one or more of these S/T-P sites inhibits, but does not completely block, Ana2's ability to be recruited to and/or maintained at centrioles. We again note that this experiment is performed in the presence of untagged WT Ana2, which probably outcompetes the mutant protein for binding to the centriole (as the mutant protein behaves as though it has been phosphorylated by Cdk/Cyclins, so its ability to incorporate into centrioles is reduced). In the absence of any WT protein, Ana2(12D/E) can presumably still localize sufficiently to centrioles to support centriole duplication in somatic cells.

## Discussion

### Centriole duplication proteins are present at surprisingly low concentrations in the embryo

Two studies have attempted to estimate the levels of one or more of the core centriole duplication proteins in human cells. Keller et al. (2014) used FCS to estimate a Sas-6 cytoplasmic concentration of ~80–360 nM, depending on the cell cycle stage, while Bauer et al. (2016) used quantitative MS to estimate the number of Plk4, Sas-6, CEP152/Asl, and STIL/Ana2 molecules in human cultured cells, which was in the ~2,000–20,000 range, ~10–15X lower than the number of γ-tubulin molecules in the cell. If the volume of a HeLa cell is ~4,000 µm³ (Zhao et al., 2008), then the concentration of these centriole proteins is in the ~1–10 nM range, which seems low, but could reflect that most somatic cells only assemble two tiny daughter centrioles during a cell cycle that can last many hours.

Given that the early *Drosophila* embryo assembles several thousand centrioles in <2 h (Foe and Alberts, 1983), we anticipated that centriole assembly proteins would be stored at higher concentrations than in somatic cells, but this does not appear to be the case. We estimate that Asl, Sas-6, Ana2, and Sas-4 are present in the ~5–20 nM range (note that 20 nM would be the concentration of the Ana2 oligomer), while the cytoplasmic

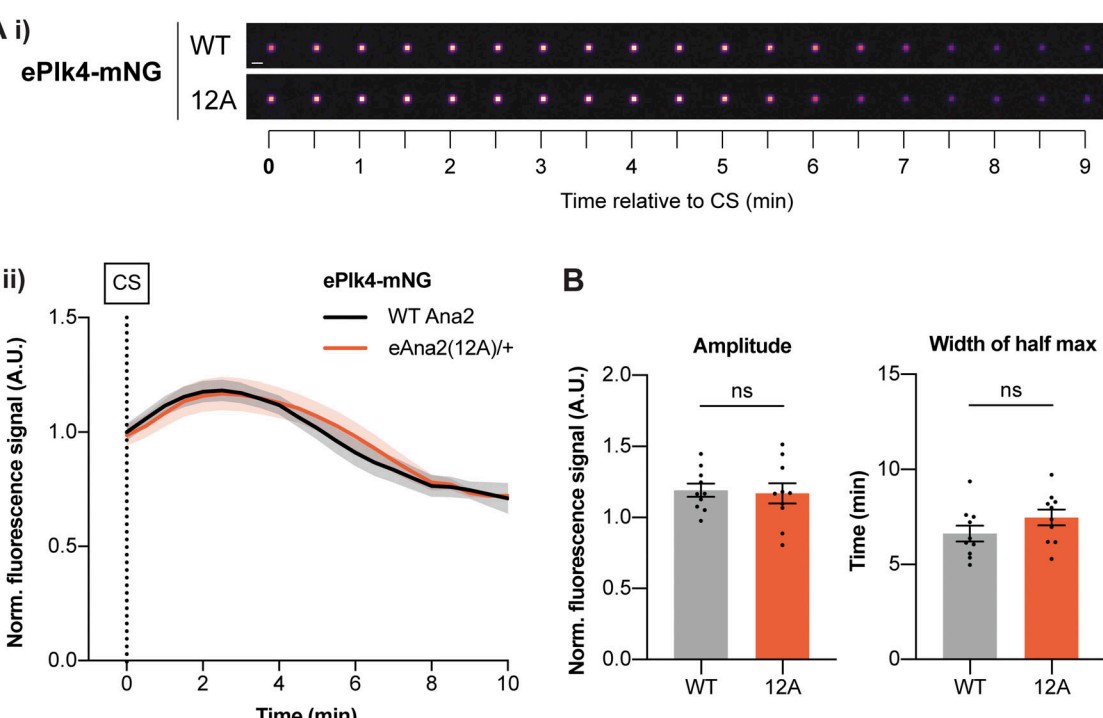

Figure 7. **The centriolar Plk4 oscillation is not dramatically perturbed in the presence of eAna2(12A). (A i)** Images show the typical centriolar recruitment dynamics of ePlk4-mNG in a WT embryo or an embryo expressing one copy of untagged eAna2(12A) in the presence of one copy of the endogenous WT *ana2* gene during nuclear cycle 12—aligned to centriole separation at the start of S-phase (CS; *t* = 0). Images were obtained by superimposing all the centrioles at each time point and averaging their fluorescence (scale bar = 1 μm). **(ii)** Graph shows the normalized (mean ± SEM) centriolar recruitment dynamics of ePlk4-mNG in the presence of either only untagged endogenous Ana2 (black) or one copy of untagged eAna2(12A) expressed in the presence of one copy of the endogenous WT *ana2* gene (orange) during nuclear cycle 12. Data was aligned to centriole separation (CS) at the start of S-phase. N = 10 embryos, *n* ~ 100 centriole pairs per embryo. **(B)** Bar charts quantify the amplitude (maximal intensity) and period (full width at half maximum intensity) (mean ± SEM) of the Plk4-mNG oscillation. Each data point represents the average value of all the centrioles measured in an individual embryo. Statistical significance was assessed using an unpaired *t* test with Welch's correction.

concentration of Plk4 is so low that we cannot measure it by FCS. Interestingly, these concentrations are similar to the MS estimates in human cell lines (Bauer et al., 2016), suggesting that the early embryo does not store a large surplus of any of these proteins. Why are these key centriole assembly proteins present at such low concentrations? Several of these proteins tend to self-assemble into larger macromolecular structures (Stevens et al., 2010b; Montenegro Gouveia et al., 2018; Kim et al., 2019; Gartenmann et al., 2020), so it seems likely that their low cytoplasmic concentration helps to ensure that they normally only start to form a cartwheel at the single kinetically favorable site on the side of the mother centriole (Lopes et al., 2015; Banterle et al., 2021). Indeed, our FCS data suggest that the concentration of Sas-6 in the embryo is low enough that it is largely monomeric in the cytoplasm, even though it is almost certainly incorporated into the centriole cartwheel as a dimer (Kitagawa et al., 2011; van Breugel et al., 2011). Storing Sas-6 as a monomer would help to ensure that it cannot spontaneously assemble into aberrant structures (Stevens et al., 2010b; Gartenmann et al., 2020), and we wonder whether storing self-assembling proteins that normally function as dimers (or higher-order homo-multimers) in cells as monomers (or lower order homo-multimers) might be a general strategy that helps to prevent their inappropriate self-assembly.

**The concentration of the core centriole duplication proteins does not change significantly during the centriole assembly process**

How cellular structures grow to the correct size is a topic of great interest (Marshall, 2015; Reber and Goehring, 2015). In the embryos of *C. elegans*, mitotic centrosome size appears to be set by a limiting cytoplasmic pool of the centrosome building block SPD-2 (Decker et al., 2011), although this does not appear to be the case for Spd-2 in early *Drosophila* embryos (Wong et al., 2022). The concept of setting organelle size with a limiting pool of building blocks is attractive, as it allows size to be controlled without the need for a specific mechanism to measure it (Goehring and Hyman, 2012). Our data, however, suggests that although the cytoplasmic concentration of the core duplication proteins is low, none of them act as limiting components to regulate centriole growth in *Drosophila* embryos. We conclude that the amount of these proteins sequestered at centrioles may be insignificant compared to the amount in the cytoplasm (a plausible scenario given the large volume of the embryo and small volume of the centriole), and/or that the rate of protein sequestration at centrioles and degradation in the embryo is finely balanced by the rate of new protein synthesis so that a constant cytoplasmic concentration is maintained.

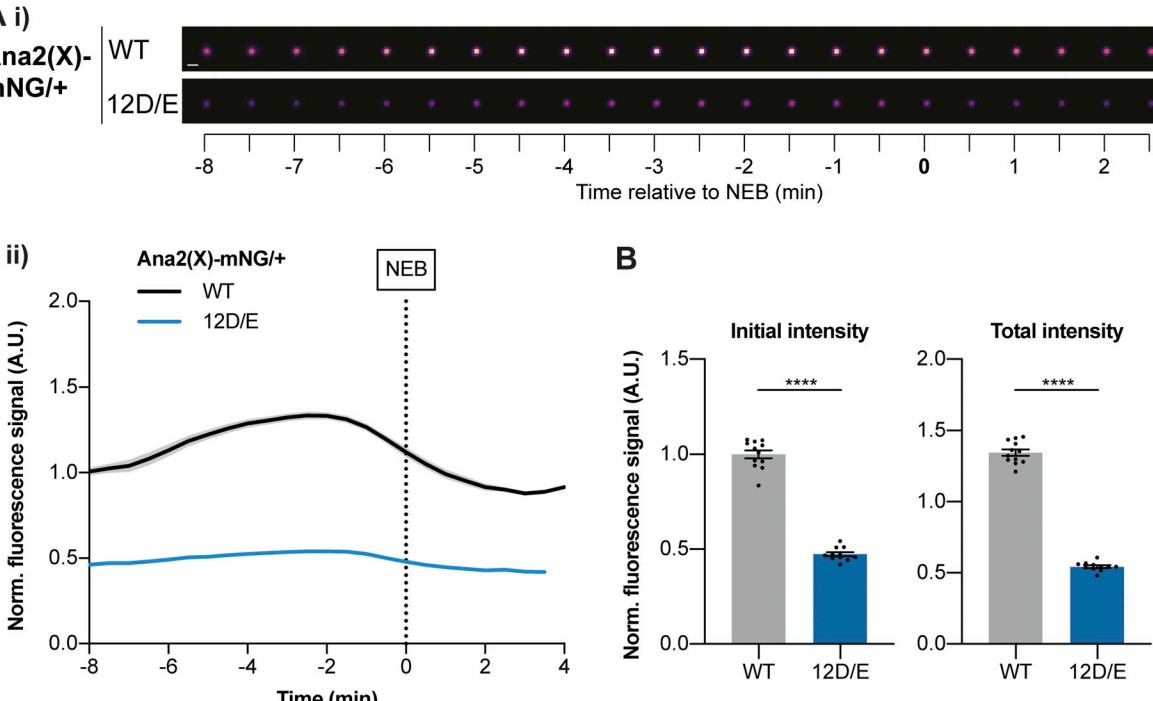

**Figure 8.** **eAna2(12D/E)-mNG is not recruited to centrioles efficiently. (A i)** Images show the typical centriolar recruitment dynamics of WT Ana2-mNG or eAna2(12D/E)-mNG in an embryo also expressing one copy of the endogenous untagged *ana2* gene during nuclear cycle 12—aligned to nuclear envelope breakdown (NEB; *t* = 0). Images were obtained by superimposing all the centrioles at each time point and averaging their fluorescence (scale bar = 1 μm). Note that the centrioles in the embryo expressing eAna2(12D/E)-mNG were very dim so their brightness has been enhanced by 2X relative to the WT control. **(ii)** Graph shows normalized (mean ± SEM) centriolar recruitment dynamics of either WT Ana2-mNG (black) or eAna2(12D/E)-mNG (blue) expressed in the presence of 1 copy of the endogenous untagged *ana2* gene during nuclear cycle 12. Data were aligned to nuclear envelope breakdown (NEB; *t* = 0). N ≥ 11 embryos, n ∼ 100–150 centriole pairs per embryo. **(B)** Bar charts quantify the normalised initial and maximal fluorescence intensity (mean ± SEM). Each data point represents the average value of all centrioles measured in an individual embryo. Statistical significance was assessed using an unpaired *t* test with Welch's correction (****, P < 0.0001).

## Cdk/Cyclin appears to phosphorylate Ana2 to modulate centriole duplication efficiency

In vertebrates, STIL binds and is phosphorylated by CDK1/Cyclin B kinase (Zitouni et al., 2016). The function of this phosphorylation is unclear, but it is thought that binding to (rather than phosphorylation by) CDK1/Cyclin B keeps STIL in an inactive state because Cdk1/Cyclin B binds to the same central coiled-coil (CC) region of STIL that binds PLK4 (Arquint et al., 2015). Our data suggest that in fly embryos Cdk1/Cyclin activity can inhibit daughter centriole growth by phosphorylating, rather than simply binding to, Ana2. Ana2's diffusion rate increases as Cdk/Cyclin activity increases toward the end of S-phase and this increase is abrogated if Ana2 cannot be phosphorylated by Cdk1/Cyclin (due to mutation of all 12 S/T-P motifs). This Ana2(12A) mutant protein can still support centriole duplication, but it is recruited to the duplicating centrioles for an unusually long period of time during S-phase (presumably because its recruitment is not efficiently inhibited by the rising levels of Cdk/Cyclin activity in the embryo), allowing the protein to accumulate at centrioles to abnormally high levels. Mutating these 12 motifs to phosphomimicking D/E motifs has the opposite effect: Ana2(12D/E) is recruited poorly to centrioles and it can no longer support the rapid cycles of centriole duplication in the early embryo. We cannot rule out that the 12A and 12D/E mutations

alter Ana2 in ways that change its conformation, multi-merization, or function in unknown ways. Nevertheless, the ability of both mutants to support centriole duplication in somatic cells and their opposing effects on Ana2's centriole recruitment are consistent with our hypothesis that these mutations prevent or mimic Ana2 phosphorylation, respectively.

*A priori*, it is perhaps surprising that the 12A and 12D/E mutants appear to support relatively normal centriole duplication in somatic cells, demonstrating that the phosphorylation of Ana2 by Cdk/Cyclins cannot be essential for duplication—although the 12D/E mutant cannot support centriole duplication in the early embryo. We speculate that while the Cdk/Cyclin-dependent phosphorylation of Ana2 reduces the efficiency of centriole duplication toward the end of the S-phase, multiple additional regulatory mechanisms—such as the oscillation in centriolar Plk4 levels (Aydogan et al., 2020; Takao et al., 2019)—help to ensure that daughter centrioles still duplicate at the right time and place even if Ana2 cannot be phosphorylated by Cdk/Cyclins. In embryos, the 12D/E mutant is lethal, as the rapidly dividing centrioles do not have time to compensate for the reduction in duplication efficiency, but this is not the case in somatic cells, where the S-phase is much longer.

### How might Ana2 phosphorylation by Cdk/Cyclins influence centriole duplication?

We do not know how the phosphorylation of Ana2 by Cdk1/Cyclins might influence centriole duplication, but we speculate that it decreases Ana2's affinity for one or more of the other core centriole duplication proteins to which it binds (e.g., Sas-6, Plk4 or Sas-4). Unfortunately, we have not been able to directly test this in vitro (as we have struggled to make well-behaved full-length recombinant proteins, possibly due to their tendency to self-assemble), and we cannot detect direct interactions between these endogenous proteins in embryo extracts, probably due to their very low cytoplasmic concentrations. Nevertheless, such a scenario would explain why Ana2's average cytoplasmic diffusion rate normally increases toward the end of the S-phase and why this increase is abrogated in the 12A mutant. Our FCS analysis also suggests that the average cytoplasmic diffusion rate of all the core duplication proteins we analyzed here increases slightly as S-phase progresses, perhaps hinting that their cytoplasmic interactions might be generally suppressed by increasing Cdk/Cyclin activity. In embryos expressing Ana2(12A), the failure to efficiently inhibit Ana2's interactions with one or more other duplication proteins toward the end of S-phase could explain why Ana2(12A) and Sas-6 can continue to incorporate into centrioles for an extended period. Such a mechanism could also explain previous observations that inhibiting Cdk1 activity can lead to centriole overduplication in flies (Vidwans et al., 2003).

Unexpectedly, expressing Ana2(12A) significantly decreased the amount of Sas-6 recruited to centrioles. This is surprising because Ana2 is thought to help recruit Sas-6 to centrioles, and centriolar Ana2(12A) levels are abnormally high. An intriguing interpretation of this finding is that while the phosphorylation of Ana2 by Cdk/Cyclins in late S-phase helps to inhibit centriole duplication, Cdk/Cyclin-dependent phosphorylation of Ana2 in early S-phase (presumably on different sites) might help promote centriole duplication by increasing the efficiency with which Ana2 interacts with Sas-6 to recruit it to centrioles. The S-phase-initiating CDK2/Cyclin kinase is required for centriole duplication (Hinchcliffe et al., 1999; Lacey et al., 1999; Meraldi et al., 1999), but its relevant substrate(s) are largely unknown. Perhaps CDK2/Cyclins phosphorylate Ana2 in early S-phase to promote centriole duplication, while CDK1/Cyclins phosphorylate Ana2 from late-S-phase onward to inhibit centriole duplication. Alternatively, the differential phosphorylation of different Cdk/Cyclin targets by different levels of Cdk/Cyclin activity plays an important part in ordering cell cycle events (Swaffer et al., 2016). Perhaps low (early-S-phase-like) levels of Cdk/Cyclin activity phosphorylate Ana2 on certain sites to promote centriole assembly, while higher levels phosphorylate Ana2 at additional sites to inhibit centriole assembly. In either scenario, Ana2 would act as a "rheostat", responding to global changes in Cdk/Cyclin activity to coordinate centriole duplication with cell cycle progression. Plk4 phosphorylates Ana2 in an ordered fashion at multiple sites to elicit sequential changes in Ana2 behavior (McLamarrah et al., 2018, 2020; Dzhindzhev et al., 2017), so it seems possible that Cdk/Cyclins might do the same.

## Materials and methods

### *Drosophila melanogaster* stocks
#### Fly stocks and husbandry

A list of all alleles and fly stocks used in this study can be found in Table S1. Flies were maintained at 18 or 25°C in plastic vials or bottles on *Drosophila* culture medium (0.68% agar, 2.5% yeast, 6.25% cornmeal [maize], 3.75% molasses, 0.42% propionic acid, 0.14% tegosept, and 0.70% ethanol). For spectroscopy/microscopy, hatching rate, and Western blotting experiments, flies were placed in egg-laying cages on fruit juice plates (40% cranberry-raspberry juice, 2% sucrose, and 1.8% agar) with a drop of yeast paste. Fly handling techniques were performed as previously described (Roberts, 1986).

#### CRISPR/Cas9-mediated fly line generation

For CRISPR/Cas9-mediated fly line generation of mNG knock-ins, a single guide RNA (gRNA) plasmid and donor plasmid for homology-directed repair (HDR) were injected into embryos expressing Cas9 from the nos promoter (BL54591) as previously described (Port et al., 2015; Port et al., 2014). The injected founder flies were crossed to balancer lines to isolate the potential knock-in allele and screened via PCR for the mNG insertion. All gRNA plasmids (pCFD3: U6:3-gRNA) were generated as described in Port et al. (2014). The gRNA target sequences were chosen based on a gRNA design algorithm to reduce potential off-target effects. Donor plasmids were assembled from individual PCR-amplified DNA fragments using NEBuilder HiFi DNA Assembly-based cloning and consisted of ∼1 kb homology arms up- and downstream of the cutting site, the mNG sequence including a short linker (N-term: 5′-TATCAAACAAGTTTGTAC AAAAAAGCAGGCTTC-3′; C-term: 5′-GACCCAGCTTTCTTGTAC AAAGTGGTTCGATATCCAGCACAGTGGCGGCCGCTCGAG-3′), and a plasmid backbone (pBluescript SK-). To prevent cleavage of the target sequence within the homology arm of the donor plasmid, point mutations in the gRNA sequence within the coding region—where possible and especially within the NGG protospacer adjacent motif (PAM) sequence—were generated that only affect individual base pairs but not the amino acid sequence of the gene. Further, the gRNA target and PAM sequences were inserted at the outer flanks of both homology arms to induce Cas9-mediated cleavage and thereby linearization of the donor plasmid in vivo.

For the generation of CRISPR/Cas9-mediated *ana2* knock-out alleles (*ana2^Δa^* and *ana2^Δb^*), two gRNAs (one for each end of the *ana2* coding region; different 5′ and -3′ gRNAs were designed to generate the two alleles) were cloned into the pCFD4 (U6:1-gRNA U6:3-gRNA) plasmid (Port et al., 2014; Port et al., 2015). The resulting plasmids were injected into BL25709 flies (y, v, nos-int; attp40) to generate gRNA-transgenic flies through attP-mediated mutagenesis. These transgenic flies were crossed to the previously described Cas9-expressing fly line BL54591 (Port et al., 2014). The *ana2^Δa^* allele (a 1,290 bp deletion that removes the entire genomic sequence between the first 15 bp and the last 9 bp of the Ana2 protein coding sequence) and the *ana2^Δb^* allele (a 1,299 bp deletion that removes the entire genomic sequence between the first 2 bp and the last 13 bp of the Ana2 protein coding sequence) were isolated

from a single founder each from the second-generation progeny.

The entire gene locus of all final knock-in and knock-out fly lines were afterward sequenced. All injections for fly line generation were performed by "The University of Cambridge Department of Genetics Fly Facility". All gRNA sequences and primers used for the generation of gRNA/donor plasmids and screening of founder flies can be found in Table S2.

### Transgenic fly line generation

Transgenic fly lines were generated via random P-element insertion (injected, mapped, and balanced by "The University of Cambridge Department of Genetics Fly Facility"). For transgene selection, the $w^+$ gene marker was included in the transformation vectors and injected into the $w^{1118}$ genetic background.

To generate Ana2 12A mutants, mutations encoding the following amino acid substitutions were introduced into an eAna2-pDONR vector encoding the genomic region of *ana2* from 2 kb upstream of the start codon up to, but not including the stop codon (Aydogan et al., 2018), using NEB Q5 site-directed mutagenesis: S63A; S84A; S101A; S172A; S257A; S284A; T301A; S345A; S348A; S365A; S395A; S403A. The resulting constructs were recombined with a destination vector encoding mNG (Aydogan et al., 2020) using Gateway technology (Thermo Fisher Scientific) to create eAna2(12A)-mNG. For untagged eAna2(12A), the endogenous *ana2* stop codon was reintroduced at its normal locus into the eAna2(12A) pDONR (described above) using NEB Q5 site-directed mutagenesis, and the resulting vector was recombined with a destination vector encoding no tag (Aydogan et al., 2018), using Gateway technology.

All other transgenic Ana2 constructs were directly cloned into the appropriate destination vector (with or without mNG tag as described above) expressed from the *ana2* core promoter (2 kb) using NEBuilder HiFi DNA Assembly. The WT Ana2 gene was amplified from genomic BL54591 DNA, and the cDNA of the two truncated Ana2 constructs (ΔCC [aa195-229], ΔSTAN [aa316-383]) was amplified from previously generated plasmids (Cottee et al., 2015). For both cDNA-containing destination vectors, Ana2's one intron was afterward reintroduced using NEBuilder HiFi DNA Assembly. The 12D/E mutations of Ana2 (S63D; S84D; S101D; S172D; S257D; S284D; T301E; S345D; S348D; S365D; S395D; S403D [S -> D and T -> E to mimic the size of the aa]) were designed *in silico* and synthesized by GENEWIZ Co. Ltd. All primers used for the generation of transgenic fly lines can be found in Table S2.

### Behavioral assays
#### Hatching experiments
Embryos were collected for 1 h and aged for 24 h at 25°C. Afterward, the hatching rate was calculated by quantifying the % of embryos that hatched out of their chorion.

#### Negative gravitaxis experiments
A negative gravitaxis assay was performed as previously described in Aydogan et al. (2018). In short, 10–15 2-d-old adult male flies in three to five technical repeats were mechanically tapped to the bottom of a measuring cylinder, and the distance

that was climbed by each individual fly within the first 5 s after the tap was measured.

### Immunoblotting and in vitro kinase assay
#### Immunoblotting
Embryos for immunoblotting were collected for 0–3 h at 25°C, chemically dechorionated, and fixed in methanol as previously described in (Stevens et al., 2010a). Afterward, the embryos were stored at 4°C at least overnight and rehydrated with 3× PBT (PBS + 0.1% Triton X-100) washes for 15 min each. Under a dissection microscope, 40 pre-cellularization stage embryos of each genotype were transferred into an Eppendorf tube with 20 µl of PBT buffer and mixed with 20 µl of 2× SDS loading dye to a final concentration of 1 embryo/µl. The samples were then lysed at 95°C for 10 min on a heat block, gently spun for a few seconds on a small lab bench centrifuge, and stored at –20°C. A total of 10 µl of the sample (which is the equivalent of 10 embryos) was loaded into each lane of a 3–8% Tris-Acetate pre-cast SDS-PAGE gel (Invitrogen, Thermo Fisher Scientific) and then transferred from the gel onto a nitrocellulose membrane (0.2 µm #162-0112; Bio-Rad) using a Bio-Rad Mini Trans-Blot system. For Western blotting, the membranes were incubated with blocking buffer (1× PBS + 4% milk powder + 0.1% Tween20) for 1 h on an orbital shaker at room temperature and then for 1 h in blocking buffer with the primary antibody (1:500 dilution). The membranes were washed 3× with TBST (TBS + 0.1% Tween-20) and then incubated for another 45 min in blocking buffer with the secondary antibody (1:3,000 dilution, horseradish peroxidase-conjugated for chemiluminescence analysis). The membranes were washed 3× for 15 min with TBST buffer, before incubation for 1 min in HRPO substrate (Thermo Fisher Scientific Super-Signal West Femto Maximum Sensitivity Substrate, #34095) at a concentration that was empirically determined for each different protein and exposed to X-ray film for ~10–600 s. The following antibodies and substrate concentrations were used: anti-Sas-6 (rabbit, [Peel et al., 2007], Substrate 1:1); anti-Ana2 (rabbit, [Stevens et al., 2010], Substrate 1:1); anti-Asl (rabbit, [Novak et al., 2014], Substrate 1:4); anti-Sas-4 (rabbit, [Novak et al., 2014], Substrate 1:3); anti-GFP (mouse, AB_390913, Substrate 1:1; Roche); anti-actin (mouse, AB_476730, Substrate 1:2; Sigma-Aldrich); anti-Cnn (rabbit, [Lucas and Raff, 2007], Substrate 1:15); anti-Gaga transcription factor (rabbit, [Raff et al., 1994], Substrate 1:5); anti-rabbit (donkey, VWR International Ltd [NA934]); and anti-mouse (sheep, VWR International Ltd [NA931-1M]).

#### In vitro kinase assay and dot blotting
Peptides for the in vitro kinase assay were synthesized by GeneScript. The complete peptide sequences were either biotin-GGAIPQFP-[S/A]-PRPHPAKK (representing the S284 site) or biotin-GGAGYRAN-[T/A]-PQAKRAKK (representing the T301 site), and for the positive control biotin-Ahx-GGAKPPKTPK-KAKKL (Ahx = aminohexanonic acid). All peptides were resuspended and stored at –80°C in 0.1 M phosphate buffer pH 7.4, 150 mM NaCl, and 2 mM DTT.

The resuspended peptides, at final concentration of 50 µM, were combined with 0.36 µg of recombinant human protein CDK1/Cyclin B (PV3292; Thermo Fisher Scientific), 1× Kinase

Buffer (#9802; Cell Signaling), 100 µM cold ATPs (#9804; Cell Signaling), and 5 µCi γ-[32P] ATP in a reaction volume of 20 µl. The reaction was incubated at 30°C for 15 min and then terminated with 10 µl of 7.5 M GuHCl. A total of 4 µl of each reaction was spotted onto a streptavidin-coated SAM2 Biotin Capture Membrane (#TB547; Promega). The membrane was air dried and then washed 2× for 30 s with 2 M NaCl, 3 × 2 min with 2 M NaCl, 4 × 2 min with 2 M NaCl + 1% $H_3PO_4$, and then 2 × 30 s with distilled water and air-dried again. The dry membrane was exposed to autoradiograph film (Carestream BioMax MR) for different lengths of time. Overnight exposures were performed at –80°C.

A loading control for the kinase assay was performed using a dot blot. About 1.2 µl of the resuspended peptide was spotted onto a nitrocellulose membrane (0.2 µm #162-0112; Bio-Rad) and left to air-dry. The dry membrane was washed in blocking buffer (PBS + 4% milk powder + 0.1% Tween-20) for 20–30 min and subsequently incubated for 45 min in Streptavidin–

HRP (Thermo Fisher Scientific) was diluted 1:3,000 in blocking buffer. The membrane was then washed 3× 10–15 min in wash buffer (TBS + 0.1% Tween-20) followed by incubation with HRPO-substrate (Thermo Scientific SuperSignal West Femto Maximum Sensitivity Substrate, #34095) for 1 min and subsequently exposed on film.

## Spectroscopy/microscopy experiments
### Embryo collection for fluorescence spectroscopy/microscopy measurements
Embryos were collected on cranberry–raspberry juice plates for 1 h at 25°C and aged at 25°C for another ~45 min. Embryos were then dechorionated by hand and mounted on a strip of glue which was positioned on either high precision 35 mm, high glass bottom µ-dishes (ibidi; for FCS/PeCoS experiments) or on MatTek dishes (1.5H thick glass bottom, MatTek Corporation). Embryos were covered in Voltalef H10S PCTFE oil (Arkema) to avoid desiccation.

### Fluorescence correlation spectroscopy (FCS) and PeCoS
Point FCS and PeCoS measurements were performed and analyzed as previously described in Aydogan et al. (2020). All measurements were conducted on a confocal Zeiss LSM 880 (Argon laser excitation at 488 nm and GaAsP photon-counting detector [491–544 nm detector range]) with Zen Black Software. A C-Apochromat 40×/1.2 W objective and a pinhole setting of 1AU was used, and spherical aberrations were corrected for on the correction collar of the objective at the beginning of each experimental day by maximizing the FCS-derived CPM value of a fluorescent dye solution. The effective volume $V_{eff}$ was previously estimated to be ~0.25 fl, determined by two independent methods: (1) comparison of the diffusion coefficient of Alexa Fluor 488 NHS Ester in water with a previously reported one (Petrášek and Schwille, 2008); (2) imaging of subresolution beads (FluoSpheres Carboxylate-Modified Microspheres, 0.1 µm). Measurements were conducted with a laser power of 6.31 µW for FCS and 10.00 µW for PeCoS, and no photobleaching was observed for any protein. The temperature of the microscope was kept between 25.0 and 26.0°C using the Zeiss inbuilt heating unit XL.

For experimental FCS recordings, consecutive cytoplasmic measurements were made 6× for 10 s each at the centrosomal

plane of the embryo. In some cases, the cytoplasmic position of the laser beam was slightly readjusted during the measurement, but the recording, in which the readjustment was made, was discarded. Erratic autocorrelation functions (usually generated when a centrosome or yolk granule moved into the point of measurement) were also discarded before all remaining curves that were fitted with eight different diffusion models in the FoCuS-point software, including one or two diffusing species with no dark state of the fluorophore, one dark state of the fluorophore (either triplet or blinking state), or two dark states of the fluorophore (triplet and blinking state; Waithe et al., 2016). The fitting boundaries were restricted to 0.4 ns–200/3,000 ms (depending on the diffusion speed of the protein), the triplet state to 1–10 µs, and the blinking state to 10–300 µs. In all models, the structural parameter AR, which denotes the ratio of the axial to radial radii (AR = $\omega_z/\omega_{xy}$) of the measurement volume, was kept constant at 5, and the anomalous subdiffusion parameter α was selected individually for each protein based on the curve's best fit (tested with 0.05 increments). The most suited model and anomalous subdiffusion parameter α were chosen based on the Bayesian information criterion (Schwarz, 1978) and were applied to all measurements of the same protein (see Table S3). After background correction and calculation of the cytoplasmic concentration, diffusion coefficient and CPM, outliers were discarded using a ROUT outlier test (applied to all 10-s recordings in GraphPad Prism [Q = 1%]). Only measurements with at least 4 × 10-s recordings were kept for further analysis. For recordings throughout an entire nuclear cycle, only embryos where all measurements fulfilled these criteria were kept. Most embryos developed at a similar speed which resulted in the same number of FCS recordings throughout the cycle, and only these embryos were used for the final analysis.

For PeCoS measurements throughout nuclear cycle 12, one continuous measurement was conducted throughout the first 9 min of the S-phase, which was then split into and analyzed as individual 60-s-long intervals. If a centriole moved through the observation spot during the measurement and caused a sharp rise in the time-trace of intensity fluctuations, the entire recording was discarded.

### Spinning-disk confocal microscopy
Embryos were imaged at room temperature using an Andor Dragonfly 505 spinning-disk system (40 µm pinholes) which was mounted on a Leica DMi8 stand (Fusion software). A 488 nm solid-state diode laser and a HCPL APO 63×/1.40 oil immersion objective were used. For the image acquisition, stacks consisting of 17 slices with a spacing of 0.5 µm in z were taken every 30 s using an Andor iXon Ultra 888 EMCCD camera.

Post-acquisition, the resulting images were first processed using Fiji (National Institutes of Health) and then further analyzed either using GraphPad Prism 8 (for Sas-6-mNG and ePlk4-mNG incorporation), methodology described in Aydogan et al. (2018) and Aydogan et al. (2020), or in a customized Python script (for Ana2-mNG [WT, 12A and 12D/E] incorporation), methodology described in Wong et al. (2022). In Fiji, the stacks were first reduced to maximum-intensity projections, which were then bleach-corrected using the exponential fit algorithm.

The background was subtracted using a rolling ball radius of 10 pixels and the centriolar pairs were tracked using the Fiji plug-in TrackMate (Tinevez et al., 2017). The following settings were chosen within TrackMate: spot diameter: 1.1 μm, no gaps between frames, only centriolar pairs that could be tracked from the beginning of nuclear cycle 12 until nuclear envelope breakdown (NEB; for Sas-6)/beginning of nuclear cycle 13 (for Ana2)/throughout the entire detection window of the oscillation (for Plk4) were kept for the final analysis.

For the Sas-6 incorporation dynamics, the regression of all centriolar pairs of each individual embryo was calculated in GraphPad Prism 8 and, in agreement with our previous studies (Aydogan et al., 2018), the "linear growth + plateau" model was the preferred model to describe centriole growth under WT conditions. Within the experiment, all dynamic curves were fitted with a "linear growth + plateau" and a "linear growth only" model and, depending on the best fit, the incorporation parameters were extracted from either of the two models. For the Plk4 incorporation dynamics, a Lorentzian model was fitted in GraphPad Prism 8 to extract the amplitude and time of the peak as previously described (Aydogan et al., 2020). For the Ana2 incorporation dynamics, the mean intensity curve from all embryos was directly displayed from the normalized raw data, and the incorporation parameters for each embryo were extracted from the initial time point and the time point with the maximum intensity. Sas-6 and Ana2 incorporation data were normalized to NEB and Plk4-mNG to centriole separation (CS) as NEB could not be identified due to the low cytoplasmic signal. The mean signal of the first time point detected under WT conditions was set as a signal of 1 and the data was normalized accordingly.

The averaged centriole images shown in the figures represent the collective behavior of all the centrioles in an embryo. They were generated by averaging the individual images of all the centrioles being tracked in an embryo at each timepoint. The images were adjusted and displayed using the same parameters for each experiment, except for Ana2(12D/E) (Fig. 8), where the intensity was doubled for optimal presentation.

### Electron microscopy
Wing-discs from 3rd instar larvae were prepared as described previously (Stevens et al., 2010). Briefly, the wing discs were dissected in PBS and then fixed in 2.5% glutaraldehyde, 4% paraformaldehyde, and 0.1% tannic acid (from a freshly prepared 10% stock) in 0.1 M PIPES buffer (pH 7.2) for 1 h (up to 2 h) at RT and left overnight in the fridge at 4°C. Samples were then washed twice in 0.1 M PIPES, followed by one wash in 50 mM glycine in 0.1 M PIPES to quench free aldehydes, and then another wash in 0.1 M PIPES. Samples were then post-fixed in 1% $OsO_4$ for 2 h at RT, followed by extensive washing in distilled water. Samples were stained with 0.5% uranyl acetate overnight at 4°C, washed in distilled water, dehydrated in an ethanol series and embedded in Agar100 (Agar Scientific). Blocks were polymerized at 50°C for 24–42 h. Semi-thin serial sections (100 nm) were obtained in a Leica EM UC7 ultramicrotome (Leica Microsystems) and stained in lead citrate. Images of centrioles in longitudinal orientation were taken on a TECNAI T12 transmission microscope (FEI) at 13,000X magnification to measure centriole length from the wing discs. The length of the MT doublets within the electron-dense area was measured using the line tool in Fiji (ImageJ).

### Data visualization and statistical analysis
All data graphs were generated, and all statistical analysis were performed in GraphPad Prism 7 or 8. The statistical tests applied to individual datasets are described in the corresponding figure legends. In general, a D'Agostino-Pearson omnibus normality test was applied to each data set to assess whether its data values resembled a Gaussian distribution. Statistical significance was defined as $P < 0.05$.

### Online supplemental material
Fig. S1 quantifies cytoplasmic FCS concentration measurements of several centriolar proteins and compares them to Western blotting experiments. Fig. S2 displays the photon-count rate per molecule (CPM) as a measure of average cytoplasmic stoichiometry for all mNG-tagged core centriole duplication proteins, both as an overall comparison and throughout the nuclear cycle 12. Fig. S3 shows a sequence alignment and the conservation of all 12 potential Cdk/Cyclin phosphorylation sites of Ana2. Fig. S4 shows that two of the most conserved potential phosphorylation sites can be phosphorylated by Cdk1/Cyclin B in vitro. Fig. S5 displays how the 12A and 12D/E mutations of Ana2 affect the flies' coordination and viability. Table S1 describes the fly stocks used in this study and in each specific experiment. Table S2 describes the primers and gRNA sequences used in this study. Table S3 describes the selected model and anomalous subdiffusion parameters (α) used for all proteins measured with FCS.

## Acknowledgments
We are grateful to Alan Wainman for help with microscopy as part of the Micron Oxford Advanced Bioimaging Unit—partly funded by a Strategic Award from the Wellcome Trust (107457). We thank members of the Raff Laboratory for advice, discussion, and for critically reading the manuscript.

The research was funded by a Wellcome Trust Senior Investigator Award (215523) to J.W. Raff, a CRUK Oxford Centre Prize DPhil Studentship (C5255/A23225), a Balliol Jason Hu Scholarship and a Clarendon Scholarship (to S.-S. Wong)., and a Medical Research Council 4-yr studentship (MR/N013468/1) to J.R. Sayers.

The authors declare no competing financial interests.

Author contributions: This study was conceptualized by T.L. Steinacker, S.-S. Wong, and J.W. Raff Investigation was done by T.L. Steinacker, S.-S. Wong, S. Saurya, Z.A. Novak, L. Gartenmann, E.J.H. van Houtum, and J.R. Sayers Computational analysis pipelines were developed by S.-S. Wong and B.C. Lagerholm. Data were analyzed by all authors. The project was supervised and administered by J.W. Raff. The manuscript was initially drafted by T.L. Steinacker, S.-S. Wong, and J.W. Raff and all authors contributed to the editing of the manuscript.

Submitted: 11 May 2022

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

# Supplemental material

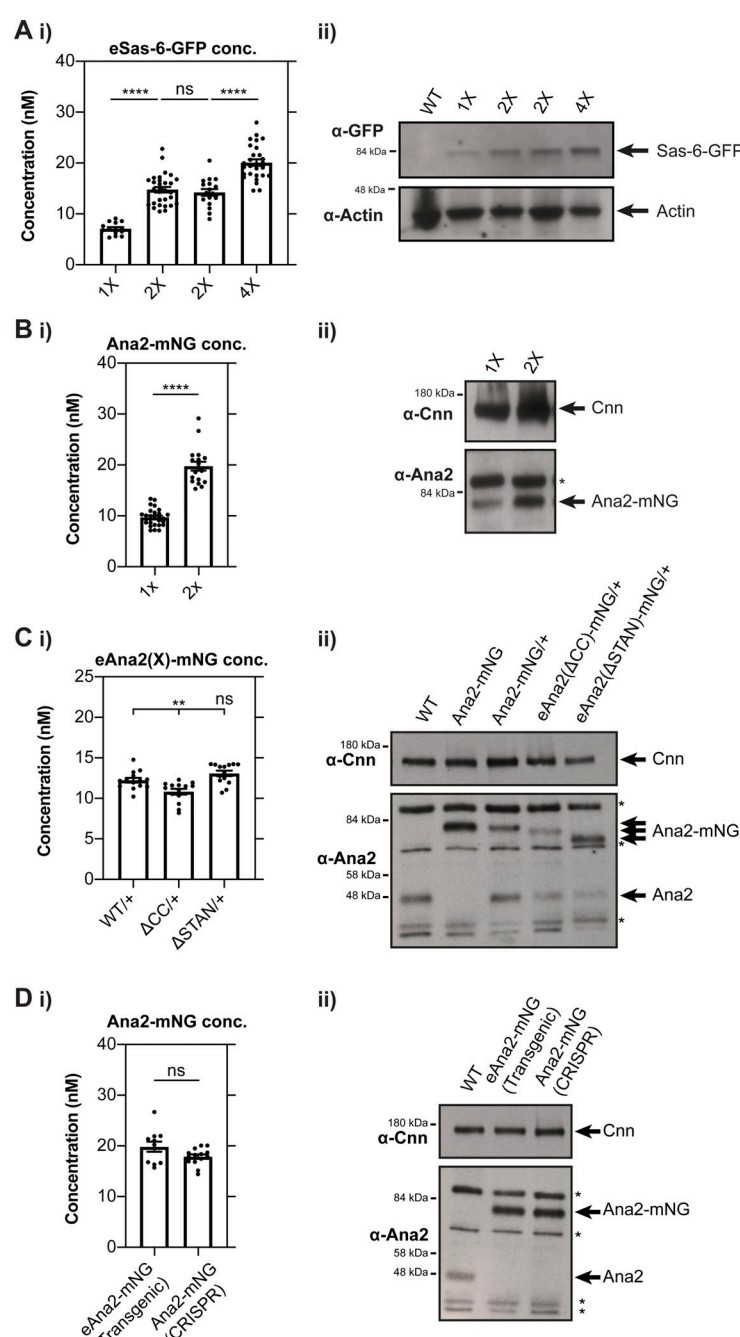

Figure S1. **FCS can be used to measure cytoplasmic protein concentrations in the early *Drosophila* embryo. (A i)** Graph shows the FCS-measured concentration (mean ± SEM) of Sas-6-GFP expressed transgenically from its endogenous promoter in embryos laid by females expressing either: 1 copy of the transgene (*1X*); 2 copies of the transgene (*2X*—shown for two different transgenic lines); four copies of the transgene (*4X*). Each data point represents the average of 4–6× 10-s recordings from an individual embryo (*N* ≥ 14). **(ii)** Western blots of 0–2 h old embryos described in A (i). **(B i)** Graph shows the FCS-measured concentration (mean ± SEM) of Ana2-mNG expressed from a CRISPR/Cas9 knock-in line as either a heterozygote (*1X*) or homozygote (*2X*). Each data point represents the average of 4–6× 10-s recordings from an individual embryo (*N* ≥ 18). **(ii)** Western blots of 0–2 h old embryos laid by the females described in B (i). **(C i)** Graph shows FCS-measured cytoplasmic concentrations (mean ± SEM) of WT Ana2-mNG, eAna2(ΔCC)-mNG and eAna2(ΔSTAN)-mNG (all in an *ana2*[+/−] heterozygous background). Each data point represents the average of 4–6× 10-s recordings from an individual embryo (*N* = 14). **(ii)** Western blots of 0–2 h embryos showing the expression levels of endogenous Ana2, a homozygous WT Ana2-mNG knock-in line, and transgenic lines expressing either WT Ana2-mNG, eAna2(ΔCC)-mNG and eAna2(ΔSTAN)-mNG (all in an *ana2*[+/−] heterozygous background). **(D i)** Graph compares FCS-measured cytoplasmic Ana2-mNG concentrations (mean ± SEM) in the transgenic WT eAna2-mNG (generated by P-element mediate transformation and expressed in an *ana2*[-/-] mutant background) and CRISPR/Cas9 knock-in Ana2-mNG lines. Each data point represents the average of 4–6× 10-s recordings from an individual embryo (*N* ≥ 11). **(ii)** Western blots of 0–2 h embryos comparing the expression levels of Ana2-mNG in the eAna2-mNG transgenic line and the Ana2-mNG knock-in line generated by CRISPR/Cas9. For Western blotting, actin or Cnn are shown as loading controls. Prominent non-specific bands are highlighted (*). A representative blot is shown from at least two technical repeats. Statistical significance was assessed using an unpaired *t* test with Welch's correction (for Gaussian-distributed data) or a Mann-Whitney test (****, P < 0.0001; **, P < 0.01).

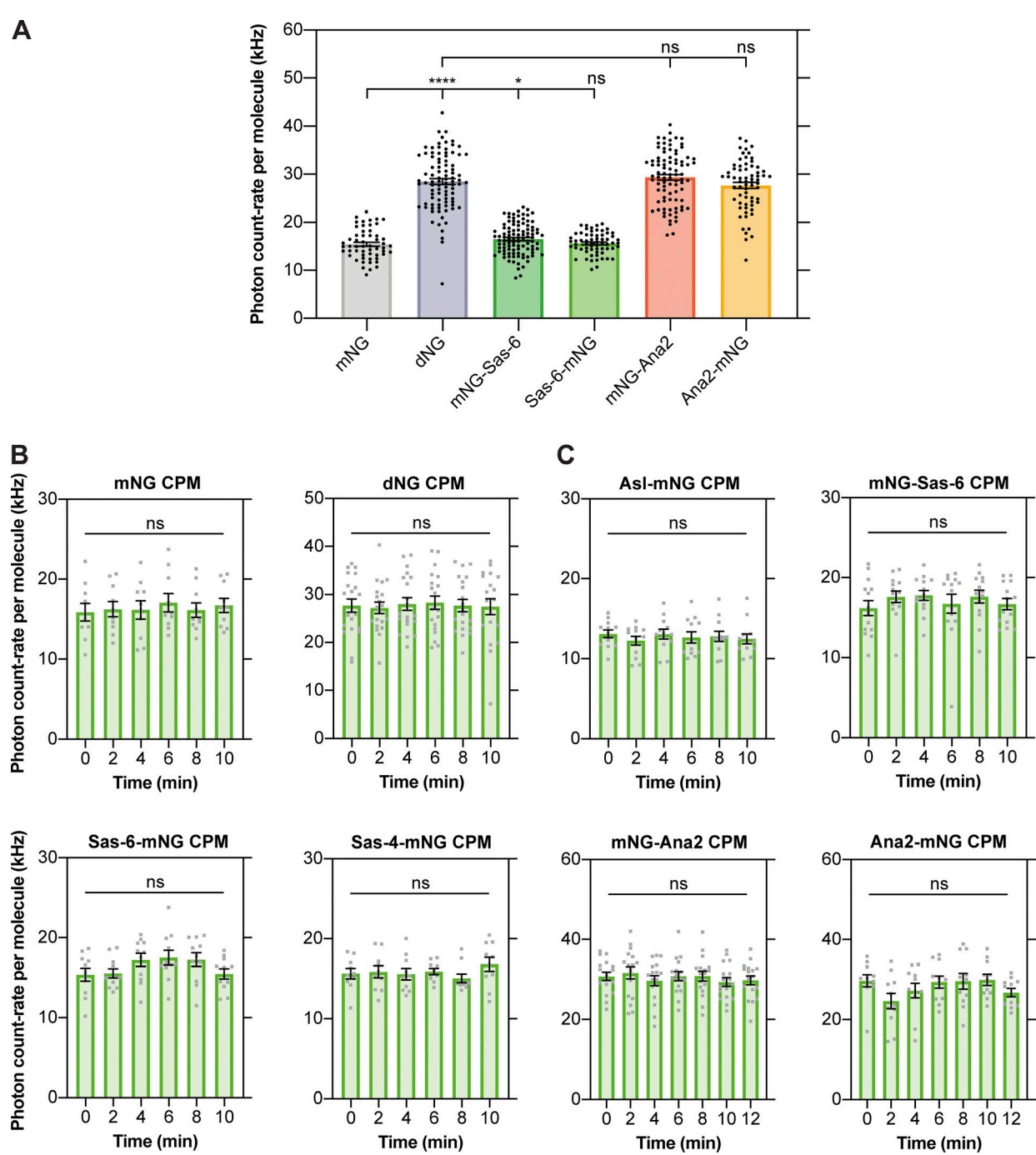

Figure S2. **Sas-6 appears to be monomeric and Ana2 multimeric in the cytoplasm, but the homo-oligomeric state of Ana2 does not appear to change during the nuclear cycle. (A)** Graph shows the average FCS-measured count-per-molecule (CPM) values (mean ± SEM) for monomeric and dimeric Neon-Green compared to mNG-Sas-6, Sas-6-mNG, mNG-Ana2 and Ana2-mNG at the beginning of nuclear cycle 12. Each data point represents the average of 4–6× 10-s recordings from an individual embryo (*N* ≥ 55). Statistical significance was assessed using an unpaired *t* test with Welch's correction (****, P < 0.0001; *, P < 0.05). **(B and C)** Graphs show cytoplasmic FCS-measured CPM values (mean ± SEM) of mNG, dNG (B) and mNG fusions to the core centriole duplication proteins (C) during nuclear cycle 12. Measurements were taken every 2 min from the start of nuclear cycle 12. Each data point represents the average of 4–6× 10-s recordings from an individual embryo (*N* ≥ 10). Statistical significance was assessed using a paired one-way ANOVA test (for Gaussian-distributed data) or a Friedman test.

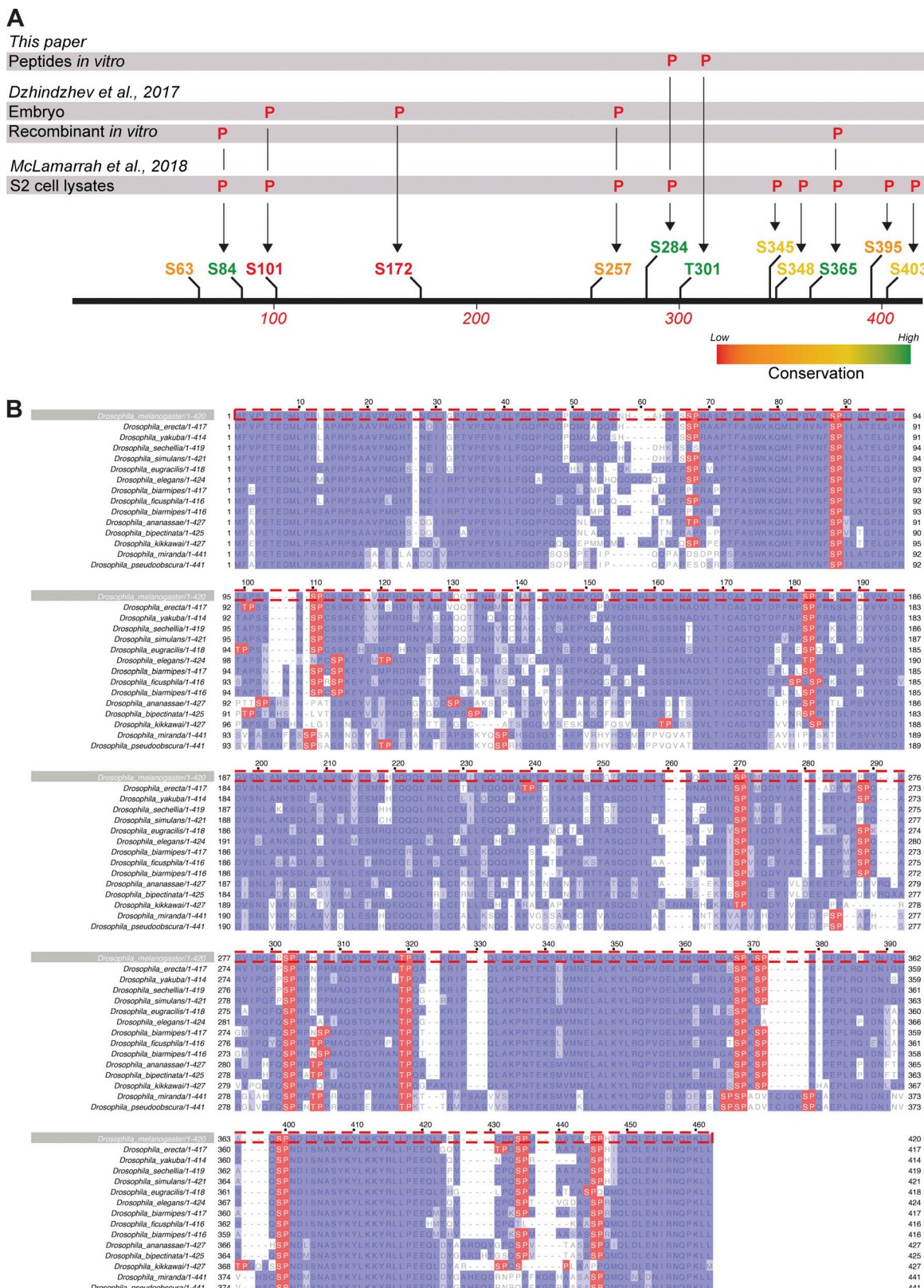

Figure S3. **There are 12 S/T-P motifs in *D. melanogaster* Ana2. (A)** Schematic illustrates the position and conservation of the S/T-P motifs in *D. melanogaster* Ana2 and indicates which of these have been shown to be phosphorylated by either Cdk/Cyclin B (this study) or a recombinant Plk4 kinase domain (Dzhindzhev et al., 2017) in vitro, or have been shown to be phosphorylated in either embryo (Dzhindzhev et al., 2017) or S2 cell extracts (McLamarrah et al., 2018) by MS. **(B)** A multiple sequence alignment (MSA) showing the conservation of S/T-P motifs (highlighted in *red*) in Ana2 from 15 different *Drosophila* species. Note that the numbering of the MSA in (B) does not precisely line up with the numbering in the schematic (A) due to the introduction of gaps in the *D. melanogaster* sequence shown in the MSA (B).

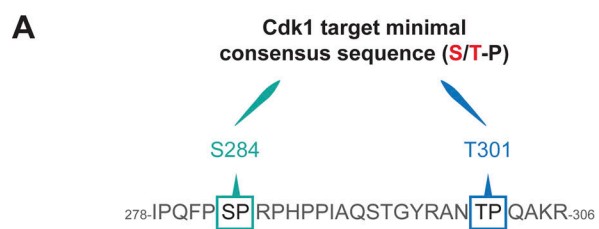

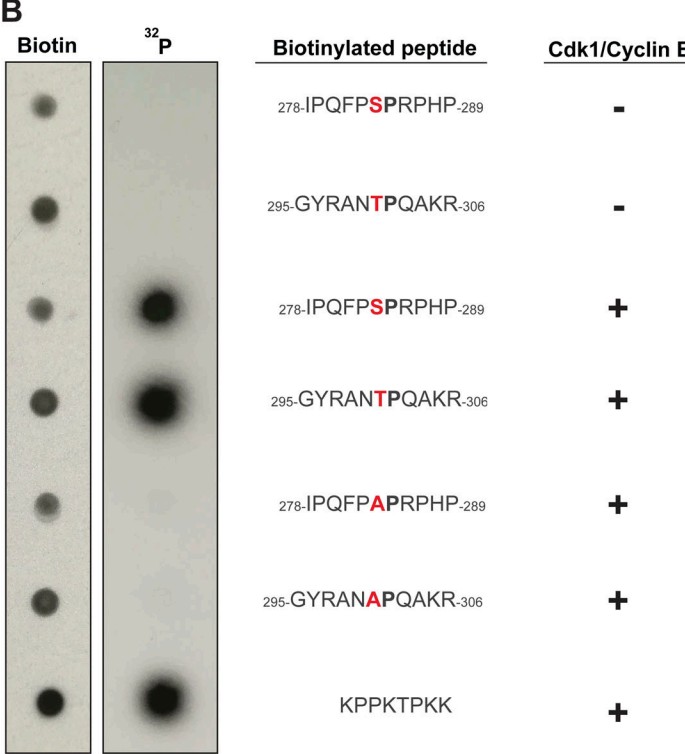

Figure S4.   **The S284 and T301 S/T-P motifs of Ana2 can be phosphorylated by recombinant Cdk1/Cyclin B kinase in vitro. (A)** The sequence of Ana2 (aa278-306) highlighting the S/T-P motifs at S284 and T301. **(B)** The indicated biotinylated peptides were synthesised in vitro and incubated with ³²P-ATP in the presence of recombinant human Cdk1/Cyclin B, or buffer alone. The reaction mixtures were spotted onto nitrocellulose membranes and autoradiographs were obtained before the membranes were probed with anti-biotin antibodies to confirm the approximately equal loading of the peptides. The peptides including S284 and T301 were phosphorylated specifically in the presence of the kinase to approximately the same extent as the positive control peptide, and this phosphorylation was essentially abolished if S284 or T301 was mutated to Alanine. We conclude that both sites are strongly and specifically phosphorylated by Cdk1/Cyclin B in vitro. A representative blot is shown from three technical repeats.

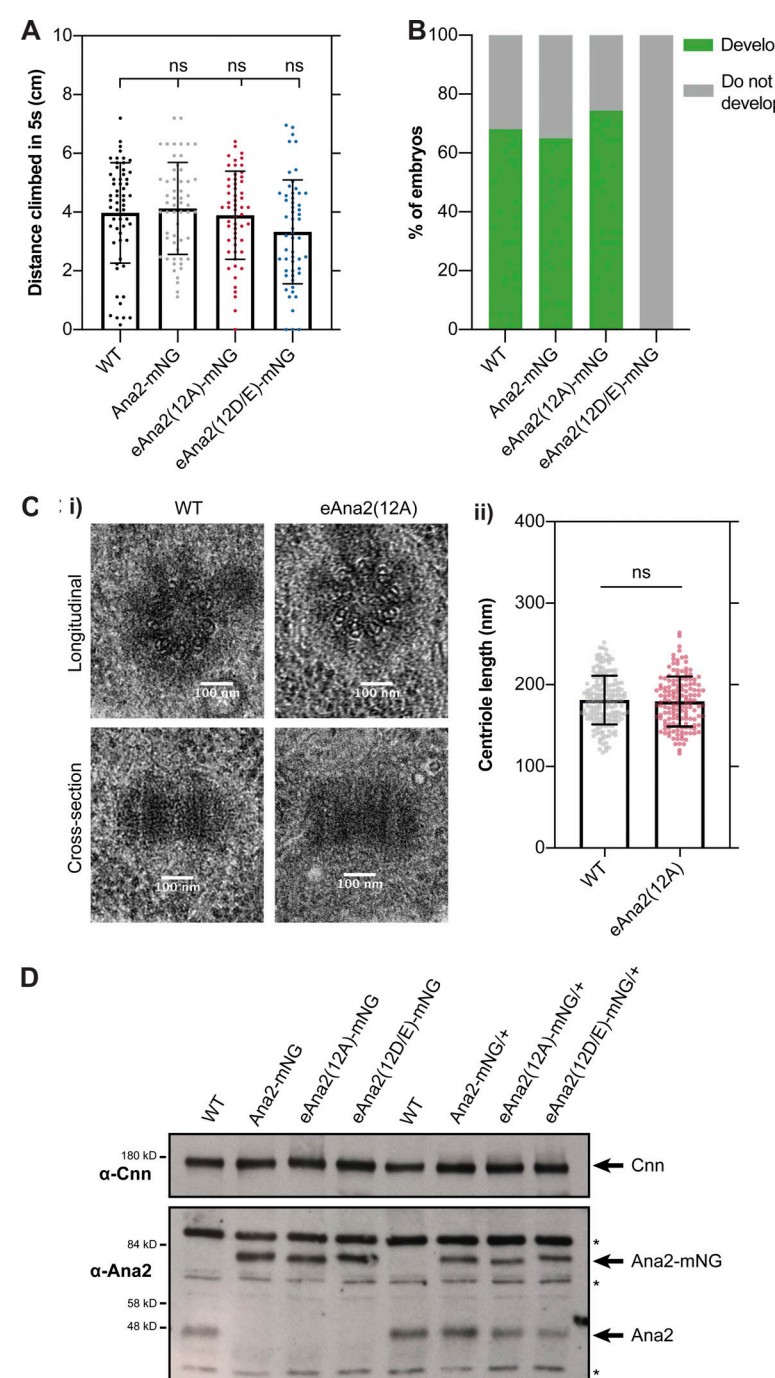

Figure S5. **The Ana2(12A) mutant appears to fully rescue the ana2⁻/⁻ mutant phenotype. (A)** Graphs quantify the distance climbed by WT or *ana2⁻/⁻* mutant flies expressing either WT Ana2-mNG, eAna2(12A)-mNG or eAna2(12D/E)-mNG in the 5 s period after all the flies have been mechanically "banged" to the bottom of a vial. This is a standard assay to measure fly coordination. Note that *ana2⁻/⁻* mutant flies are completely uncoordinated, so they cannot climb any distance at all. All three alleles, WT, 12A and 12D/E rescue this phenotype, suggesting that centriole duplication and cilia formation are unperturbed in these "rescued" flies. Each individual point on the graph represents the average distance climbed by a single fly in an individual experiment. 10–15 flies were measured in 4–6 technical repeats for each genotype. Statistical significance was assessed using an unpaired *t* test with Welch's correction. **(B)** Graph quantifies the percentage of embryos that hatch as larvae when laid by either WT females or *ana2⁻/⁻* mutant females expressing either WT Ana2-mNG, eAna2(12A)-mNG or eAna2(12D/E)-mNG. Note that these experiments were conducted when the laboratory was experiencing a general problem with Fly food, whereby many of our laboratory strains were laying embryos that did not hatch at their normal high frequencies (usually >85% for WT controls); ~400 embryos were counted for each genotype. **(C i)** EM Images show exemplar centrioles in either WT or *ana2⁻/⁻* mutant expressing eAna2(12A) 3rd instar larval wing discs. We examined a total of ~150 centrioles from five wing-discs of each genotype and identified no obvious morphological defects. **(ii)** Graph shows centriole length—scored blind in longitudinal EM sections, as depicted in the bottom panels in C (i)—in *ana2⁻/⁻* mutant 3rd instar larval wing discs expressing either WT Ana2 or eAna2(12A). Statistical significance was assessed using an unpaired *t* test with Welch's correction. **(D)** Western blots of 0–2 h embryos comparing the expression levels of Ana2-mNG, eAna2(12A)-mNG and eAna2(12D/E)-mNG. Prominent non-specific bands are highlighted (*). Cnn is shown as a loading control, and a representative blot is shown from at least two technical repeats.

**Provided online are Table S1, Table S2, and Table S3. Table S1 shows alleles and fly stocks used in this study. Table S2 shows primers and gRNA sequences used in this study. Table S3 shows the selected model and anomalous subdiffusion parameter α for all proteins measured with FCS.**

