## [Peer Review File · The Journal of Cell Biology]

Centriole growth is limited by the Cdk/Cyclin-dependent phosphorylation of Ana2/STIL

Thomas Steinacker, Siu-Shing Wong, Zsofia Novak, Saroj Saurya, Lisa Gartenmann, Eline van Houtum, Judy Sayers, B Lagerholm, and Jordan Raff

Corresponding Author(s): Jordan Raff, University of Oxford

Review Timeline:

Submission Date:	2022-05-11
Editorial Decision:	2022-06-13
Revision Received:	2022-06-29

Monitoring Editor: Arshad Desai

Scientific Editor: Andrea Marat

Transaction Report:

DOI: <https://doi.org/10.1083/jcb.202205058>

Revision 0

Review #1

1. Evidence, reproducibility and clarity:

Evidence, reproducibility and clarity (Required)

Centriole duplication is a conserved pathway that need to be tightly regulated. The key enzyme of centriole assembly is Plk4 which is recruited to the centrioles and undergoes dynamic re-localization from a ring-like pattern around a centriole to a dot-like morphology at the daughter centriole assembly site. This event is central for inducing centriole biogenesis. Plk4 then phosphorylates Ana2/STIL which allows recruitment of Sas-6 to form the cartwheel structure for centriole assembly.

In the present study, Steinacker, Wong et al. monitor how cytoplasmic concentrations of the key proteins in centriole assembly, Plk4, Asl/Cep152, Ana2/STIL, Sas-6 and Sas-4/CPAP change during the centriole assembly process in the *Drosophila* embryo by using fluorescence correlation spectroscopy (FCS) and Peak Counting Spectroscopy (PeCoS). They find that their concentrations remain constant with exception of Ana2/STIL of which cytoplasmic diffusion rate increased at the end of S-phase and is dependent on phosphorylation by Cdk1/CyclinB. Phosphorylated Ana2/STIL blocks centriole duplication thus preventing premature initiation of centriole duplication in mitosis.

Major comments

The manuscript is interesting and very well written. Most of the experiments are carefully performed. However, there are some important aspects for improvements that are listed below

Additional experiments:

- Figure 3: the transgenic flies that were generated here, Δ CC and Δ STAN, still contain wild-type Ana2. So, the authors therefore need remove or dampen their claim that the change in Ana2's cytoplasmic diffusion does not depend on its interaction with Sas-6 (page 11).
- Figure 5A: is the observed reduced recruitment of Sas-6 by Ana2(12A) due to a decrease in binding affinity? This should also be shown by analyzing protein-protein interactions between Ana2(12A) and Sas-6 biochemically.
- The authors use an Ana2(12A) mutant which comprises putative Cdk1 phosphorylation sites that have been identified in Mc Lamarrah et al. JCB 2018. However, only three of them were phosphorylated by Cdk1/cyclin B in vitro (Fig. S6). Are all these 12 putative Cdk1 phosphorylation sites important in vivo? Did the authors generate the Ana2(3A) or the S284A/T301A mutants to see whether it can rescue the ana2^{-/-} mutant phenotype similar to the 12A mutant? These might be sufficient to observe the phenotype.

- Figure 6: is the interaction between Plk4 and Ana2(12A) impaired? Similarly, Plk4 activity and phosphorylation of Ana2(12A) by Plk4
- Figure 7: Phosphomimetics, in this case 12 amino acid changes, have the disadvantage of introducing more negative charge than the phosphorylated residue. The Ana2/(12D/E)-mNG is not efficiently recruited to centrioles. Is effect also observed for the Ana2/(3D/E) mutant?

****Minor comments****

Figure S1: only mNG-tagged centriolar proteins are shown. An empty mNGtag or an mNG-tagged non-centriolar protein should be shown to exclude that the tag by itself shows centriolar localization or somehow affects the localization

S4C: Sas6-mNG CPM error bars are missing for the 10min time point

S5A: What are the expression levels of the Ana2(12A) mutant? The expression levels shown in this Figure are not similar.

2. Significance:

Significance (Required)

Centriole duplication normally begins at the G1/S phase transition. An important question in the field is how premature centriole duplication in mitosis is prevented. The authors used fluorescence correlation spectroscopy (FCS) and Peak Counting Spectroscopy (PeCoS) to study the major conserved proteins in the centriole assembly pathway and found that only Ana2/STIL's cytoplasmic diffusion increases at the end of S-phase. It is known from the literature that Cdk1 prevent Plk4-STIL complex assembly in centriole biogenesis by directly competing with Plk4 for the CC domain of Ana2/STIL (Zitouni et al. *Curr Biol* 26, 1127-1137 (2016). However, Ana2/STIL can also bind to Plk4 via its conserved C-terminal region of STIL (Ohta et al., *Cell Reports* 11, 2018; McLamarrah et al., *J Cell Biol* 2018, 217, 1217-1231). The work by Steinacker, Wong et al. suggest that at least in fly embryos, growth of the daughter centriole is regulated though phosphorylation of Ana2 by Cdk1/CyclinB rather than binding. The findings described in this manuscript are interesting for a broad range of scientists from both the centrosome and mitosis fields

Expertise of the reviewer: centriole biogenesis, structural and numerical centrosomal aberrations in disease

3. How much time do you estimate the authors will need to complete the suggested revisions:

Estimated time to Complete Revisions (Required)

(Decision Recommendation)

Between 1 and 3 months

4. Review Commons values the work of reviewers and encourages them to get credit for their work. Select 'Yes' below to register your reviewing activity at Publons; note that the content of your review will not be visible on Publons.

Reviewer Publons

Yes

Review #2

1. Evidence, reproducibility and clarity:

Evidence, reproducibility and clarity (Required)

Centriole growth is not limited by a finite pool of components, but is limited by the Cdk1/Cyclin-dependent phosphorylation of Ana2/STIL

Authors: Thomas L. Steinacker, Siu-Shing Wong, Zsofia A. Novak, Saroj Saurya, Lisa Gartenmann, Eline J.H. van Houtum, Judith R. Sayers, B. Christoffer Lagerholm, Jordan W. Raff

Centriole biogenesis is a tightly regulated process that occurs once per cell cycle. Defects in this process can lead to the acquisition of abnormal centriole numbers which has been linked to several human diseases. Centriole duplication starts with the assembly of a procentriole on the mother centriole in early S-phase followed by procentriole growth during G2 phase. A big question in the centrosome field is how new procentrioles assemble at the right time and acquire the correct final size.

In this manuscript, Wong et al. analyse whether the cytoplasmic concentration of several proteins changes during centriole assembly (Asl, Plk4, Ana2, Sas-6, and Sas-4). The authors show that the cytoplasmic concentration of these proteins remains constant during centriole duplication, indicating they are not limiting components for procentriole assembly. Nevertheless, the authors found that Ana2/STIL's cytoplasmic diffusion rate increases before the onset of mitosis, concurrent with an increase in Cdk1/Cyclin activity. Mutation of 10 putative phosphorylation sites in Ana2 prevented the diffusion rate change and enabled centrioles to grow for a longer period. This suggests that phosphorylation of Ana2/STIL by Cdk1/Cyclin could control the period of centriole growth.

****Minor points:****

In the introduction, the authors describe how PLK4 is required to recruit STIL and Sas-6 to promote the formation of the cartwheel during centriole duplication. However, there is also literature describing a role for STIL in regulating PLK4 abundance and localization pattern (i.e ring or dot) at the centriole.

The authors note that the levels of the Ana2(12A) mutant keep increasing until the onset of mitosis. The authors claim that this phenotype is consistent with the timing of increased Cdk1 activity. It would be interesting to show the increase in Cdk1 kinase activity over the same time-course and test whether dampening Cdk1 has the same effect on Ana2 recruitment.

While I appreciate detecting in vivo phosphorylation sites can be very challenging, It would be valuable to show the 10 Ana2 phosphorylation sites can be phosphorylated by Cdk1, at least in vitro.

****Other points:****

Figure 3: Amino acids numbers for CC domain are not the same in the figure and in the figure legend.

Figure 5Aii, the x-axis should be changed to minutes for easier comparison with other figures.

There are some typos in the figure legends.

2. Significance:

Significance (Required)

This study attempts to address a central question in the centrosome field: how centriole growth is controlled. Although the paper does not provide a detailed mechanistic advance, the authors do provide some evidence against a limited pool of centriole components controlling centriole length, and they are careful not to overstate conclusions. The manuscript is well written and easy to follow. While it is not clear at present how phosphorylation of Ana2 alters its diffusion rate or limits centriole growth, I feel the study will be of interest to members of the centriole community and will stimulate new lines of investigation. Given that the cartwheel stops elongating in S phase in mammalian systems, it is not clear if the mechanism proposed would be conserved. That notwithstanding, I found this to be a rigorous study that advances our understanding of the regulation of the centriole duplication.

3. How much time do you estimate the authors will need to complete the suggested revisions:

Estimated time to Complete Revisions (Required)

(Decision Recommendation)

Less than 1 month

4. Review Commons values the work of reviewers and encourages them to get credit for their work. Select 'Yes' below to register your reviewing activity at Publons; note that the content of your review will not be visible on Publons.

Reviewer Publons

Yes

Review #3

1. Evidence, reproducibility and clarity:

Evidence, reproducibility and clarity (Required)

The manuscript entitled "Centriole growth is not limited by a finite pool of components, but is limited by the Cdk1/Cyclin-dependent phosphorylation of Ana2/STIL" by Steinacker et al. nicely demonstrates that centriole growth in *Drosophila* embryos is not limited by a finite pool of core centriole components as in other systems. In contrast, they unveiled a specific elevated cytoplasmic diffusion rate of Ana2/STIL towards the end of the S-phase, correlating with the rise of Cdk1/Cyclin activity, that they hypothesize is important for the abrupt stop of centriole growth before mitosis (end of S phase). They found using an Ana2 mutant (12A) that cannot be phosphorylated by Cdk1/Cyclin that this elevated diffusion rate is abrogated, demonstrating that this kinase is involved in this process. The authors further conclude that daughter centrioles grow at a slower rate for an extended period (as followed by SAS-6 incorporation at centrioles in the context of the 12A mutant). Thus, the authors conclude that this novel mechanism ensures that daughter centrioles stop growing at the correct time and propose that it could be part of the explanation why centriole duplication does not occur during mitosis.

Overall, this is a solid study that is well written and easy to follow. The text and figures are well presented and the quality of the data is convincing. This manuscript would be of great interest not only to the centrosome field but also more generally to cell biologists.

I do not have major concerns regarding the experiments. However, I would like to propose some minor comments/clarification in order to further improve the manuscript.

****Suggestions for additional improvements:****

My main comments are related to the phosphorylated mutants of Ana2 (12A) and (12D/E).

1. To study the impact of Cdk1 on Ana2, the authors generated a mutant where 12 potential Cdk1 sites have been replaced by Alanine (12A). Although I acknowledge that all controls were properly done on this mutant and that the 12A protein is functional since it rescues the *ana2*^{-/-} mutant phenotype, one can still wonder whether this could not affect somehow the overall protein conformation, or structure. Maybe this could simply be stated somewhere in the manuscript.

2. The authors mentioned that there is evidence that 10 Cdk1 sites in Ana2 are phosphorylated in vivo and they further demonstrate convincingly that 2 of the most conserved sites can be phosphorylated in vitro by Cdk1/CyclinB (Figure S6). Could the authors include the alignment showing the potential phosphorylation residues and highlight the 12 that were mutated and show the overall conservation of these sites? It would be easier to find the residues as from the scheme of Fig. 3A it is not easy to find which residues are mutated (although the information can be found in the method section p. 32).

3. My main confusion regarding the phosphorylation mutants 12A or 12D/E comes from the fact that both can rescue the *ana2*^{-/-} mutant phenotype, which indicates that the mutant protein is functional and that somehow these sites are not fully important for centriole duplication or are not solely responsible for this type of regulation. Is this interpretation correct, this is somehow what I take from the end of the discussion p.21? if true maybe it should be a bit more emphasized.

4. Moreover, I would have expected since centriole growth does not stop abruptly (one could talk about "prolonged" centriole growth) in the 12A mutant that centrioles would be longer. However, this is not the case as shown in Figure S7. One possible explanation would be that even though the centriole growth is extended (looking at SAS6 as a proxy), the slope/rate of incorporation is lower. Could you please comment on this more? I think this is an important point of discussion/interpretation of the results.

5. The authors nicely show that the 12A mutant, despite similar expression levels as the tagged Ana2^{WT}, continued to accumulate at centrioles till NEBD (consistent with the hypothesis that Cdk1 cannot phosphorylate it and thus stops its recruitment). But how can the 12A levels decline at centrioles in mitosis where Cdk1 activity is the highest? This would mean that Cdk1 activity/level regulates Ana2 differentially over time or that other mechanisms might be at play. The authors mention in the discussion the attractive hypothesis of the "rheostat" (p. 20) but maybe a further discussion on an alternative mechanism could be also interesting.

Could the fact that the 12A level decreases in mitosis also explain the lack of centriole phenotype if we would imagine that levels at centrioles would stay high? Could the authors comment on this? They mention it briefly (p.14 and p.21) but if they could expand a bit would be great.

6. I was a bit confused about how the 12D/E mutant that is not recruited efficiently to centrioles could rescue the *ana2*^{-/-} mutant centrioles? Could the authors comment on this, please?

7. p.16 still about the 12D/E that is not properly recruited to centrioles even in presence of one WT copy of Ana2 (untagged): The authors conclude that "phosphorylation at one or more of these S/T sites inhibits, but does not completely block, Ana2 recruitment to and/or maintenance at centrioles". Could the mutation also prevent Ana2 homo-oligomerization? In other words,

could this result suggest that the 12D/E cannot interact with untagged Ana2WT and be recruited to centrioles? Is it a possibility?

2. Significance:

Significance (Required)

In this manuscript, the authors address two major fundamental questions:

1. the mechanism that restricts strict cell cycle regulation of centriole duplication
2. How daughter centrioles grow to the correct size.

These questions are very important and this study provides some clues on the mechanisms that can be at play, among which Cdk1/cyclin seems to be involved.

In addition, this paper raises an interesting point in showing that the core centriole duplication components concentration is as low in human cells as in fast-dividing *Drosophila* embryos in the range of 5-20nM. This is very interesting as it was commonly thought that embryos would have a stockpile of core components to ensure fast and numerous centriole duplication cycles. Furthermore, they found that these concentrations remained constant using FCS or Pecos, demonstrating that core centriole components concentrations are not rate-limiting for centriole duplication (over time) in this system. Instead, they propose an alternative hypothesis whereby Cdk1/Cyclin phosphorylation of Ana2/STIL would be important to regulate centriole growth and ensured timely duplication (ie no duplication in mitosis, when Cdk1 activity is high).

In this context, this study would certainly have a broad interest and impact on cell biologists.

Reviewer's expertise:

Centrioles, microtubules, microscopy, cell biology.

3. How much time do you estimate the authors will need to complete the suggested revisions:

Estimated time to Complete Revisions (Required)

(Decision Recommendation)

Less than 1 month

4. Review Commons values the work of reviewers and encourages them to get credit for their work. Select 'Yes' below to register your reviewing activity at Publons; note

that the content of your review will not be visible on Publons.

Reviewer Publons

No

Manuscript number: RC-2022-01297

Corresponding author(s): Jordan, Raff

1. General Statements [optional]

We would like to thank the reviewers for their helpful and constructive comments.

Reviewer #1

This reviewer thought our findings would be of interest to a broad range of scientists from both the centrosome and mitosis fields, but noted some important aspects for improvements.

Additional Experiments (we number these points for ease of discussion).

1. *Figure 3. The reviewer points out that because our analysis of Ana2- Δ CC and Ana2- Δ STAN mutant proteins was conducted in the presence of endogenous WT protein, we should be more cautious in our interpretation.* We agree and apologise for overstating these findings. We have now rewritten the title and text of this section to be more cautious (p11, para.2)

2. *Figure 5A. The reviewer wonders whether the reduced recruitment of Sas-6 in the presence of Ana2(12A) is due to reduced binding, and they request we test this biochemically.* This is our favoured interpretation, but we have been unable to test this biochemically for two reasons. First, although we have successfully purified several recombinant Sas-6 and/or Ana2 fragments (Cottee et al., *eLife*, 2015), the full-length proteins are poorly behaved (tending to precipitate, likely due to their inherent ability to self-oligomerise). Thus, we have been unable to reconstitute their interaction *in vitro*. Second, as we show here, the proteins are normally expressed in embryos at surprisingly low concentrations (~5-20nM), and we can detect no interaction between them in coimmunoprecipitation experiments from embryo extracts (not shown). Indeed, this concentration is so low that Sas-6 does not even appear to form a homo-dimer in the embryo, even though Sas-6 clearly functions as a homo-dimer in centriole assembly (new Figure S4A). We now explain these points, and state that our favoured hypothesis that Ana2(12A) has reduced affinity for Sas-6 (or other core duplication proteins) remains to be tested (p22, para.2).

3. *The Reviewer wonders if all 12 of the potential Cdk1 phosphorylation sites that we mutate in Ana2(12A) are important in vivo, and whether we have tested whether mutating fewer sites (e.g. the two sites [S284/T301] that we show are phosphorylated by Cdk1/Cyclin B in vitro) might be sufficient to recapitulate the Ana2(12A) phenotype.* We have now tested this by mutating just the S284/T301 sites to Alanine [Ana2(2A)], but the results were not very informative (Reviewer Figure 1 [RF1]). Whereas Ana2(12A) is recruited to centrioles for a longer period and to higher levels than WT Ana2 (Figure 4A), Ana2(2A) is recruited to centrioles for a normal period but to lower levels (RF1A,B). The interpretation of this result is complicated because western blots show that

Ana2(2A) is also present at lower-levels than normal (RF1B). Thus, it is clear that Ana2(2A) does not recapitulate well the behaviour of Ana2(12A). We have decided not to present this data as it is difficult to interpret and it does not change any of our conclusions.

4. *Figure 6.* The reviewer asks whether the 12A mutations impair the interaction with Plk4, influence Plk4's kinase activity or the ability of Plk4 to phosphorylate Ana2. These are excellent questions but, for the same reasons described in point 2 above, we cannot address them biochemically as we cannot purify well-behaved recombinant full-length Ana2 or active Plk4 *in vitro*, and both proteins are present at such low levels in the embryo that we cannot detect any interaction between them in embryo extracts. We are working hard to reconstitute *in vitro* systems to probe these important points, but it may be sometime before we are able to do so.

5. *Figure 7.* The reviewer suggests that the 12D/E phosphomimetic substitutions introduce more negative charge than the putative phosphorylation of Ser/Thr residues and they ask if the Ana2(2D/E) [stated as Ana2(3D/E)] is, like the Ana2(12D/E) mutant, not efficiently recruited to centrioles. This is a fair comment, but we have not analysed an Ana2(2D/E) mutant because, as described in point 3 above, the Ana2(2A) mutant did not recapitulate well the Ana2(12A) phenotype.

Minor comments

1. *Figure S1.* The reviewer requests that we show that the mNG tag on its own is not recruited to centrioles. We do not show this (as it would create a lot of white space in this Figure), but now state that mNG and dNG do not detectably localise to centrioles (p7, para.1).

2. *Figure S4C.* We have included the missing error bars (now Figure S4B).

3. *Figure S5A.* The reviewer asks about the expression levels of the Ana2(12A) mutant, which are not shown in this Figure. They also state that the expression levels of the transgenes shown in Figure 5A are not similar. The expression level of Ana2(12A) is shown in Figure S9, as this data was analysed independently of the other mutant proteins shown in Figure S5. We agree that it was overly simplifying the situation to state that the expression levels of WT Ana2-mNG, eAna2(Δ CC)-mNG and eAna2(Δ STAN)-mNG were "similar" (Figure S5), and we now specifically mention the differences between them (p11, para.3).

Reviewer #2

This reviewer found this a rigorous study that advances our understanding of the regulation of centriole duplication, but raised some minor points.

Minor Points

The reviewer requests that we mention the literature describing how Ana2/STIL can influence the abundance and centriolar localisation of Plk4. We apologise for this omission, and have amended our description of this literature in the Introduction to include this point (p3, para.2).

*The reviewer notes that we interpret the ability of the Ana2(12A) mutant to keep incorporating into the centrioles for a longer period as being consistent with our idea that rising levels of Cdk activity during S-phase normally reduce the ability of WT Ana2 to bind to the centriole. They ask us to show how Cdk activity increases over this time-course, and to test whether dampening Cdk has the same effect on Ana2 recruitment (i.e. allows Ana2 to be recruited for a longer period). The time-course of Cdk activation in these embryos has been reported previously (Deneke et al., *Dev. Cell*, 2016; we present the relevant data from this paper in RF#2A [black line]). This reveals how Cdk activity rises throughout S-phase, which is crucial for our model. To assess the effect of dampening Cdk activity in these embryos we have now analysed the effect of halving the genetic dose of Cyclin B (RF#2B). This perturbation extends S-phase length, but has a complicated effect on the recruitment dynamics of Ana2 (RF#2B). As we would predict, Ana2 is recruited to centrioles for a longer period in these embryos, but it is also recruited more slowly (so it accumulates to lower levels). This is consistent with our hypothesis that Cdk1 activity might first stimulate and then ultimately inhibit the centriolar recruitment of Ana2. The interpretation of this experiment is not straightforward, however, as dampening Cdk1 activity alters Ana2 recruitment dynamics (and many other processes in the embryo) in complicated ways, so we have decided not to include it in the manuscript.*

The reviewer suggests that it would be valuable to show that all 12 of the potential Cdk1 phosphorylation sites in Ana2 can be phosphorylated by Cdk1 in vitro. We think this would not be particularly informative as our hypothesis does not rely on all 12 sites being phosphorylated to generate the Ana2(12A) phenotype. We simply mutate all 12 sites because we don't know which, if any, are relevant. Thus, showing that some/all of the 12 sites can/cannot be phosphorylated in vitro does not test any hypothesis and would not change any of our conclusions. We now explain our thinking on this in more detail (p12, para.2)

Other points

Figure 3. We have corrected the amino-acid numbering mistakes.

Figure 5Aii. We have changed the x-axis (time) labelling in this and all other Figures.

Figure Legends. We have tried to eliminate the typos from the Figure legends, and apologise that these errors made it through to the final submitted version of our manuscript.

Reviewer #3

This reviewer thought our manuscript would be of great interest to not only the centrosome field but also to cell biologists more generally. Although they had no major concerns, they made a number of suggestions for improvements.

1. As the reviewer suggests, we now explicitly state that although the Ana2(12A) mutant appears to be largely functional, the overall conformation of the protein may be altered, changing its function in ways we do not appreciate (p21, para.2).

2. *The reviewer suggests we include a multiple sequence alignment of Ana2/STIL proteins to provide more context about the distribution and conservation of the 12 S/T-P sites mutated in Ana2(12A).* This is an excellent idea, and we now include this in a new Figure S6, where we also provide more information about which of these sites have been shown to be phosphorylated in embryo or S2-cell extracts

3. *The reviewer is confused as to why the 12A and 12D/E mutants rescue the ana2^{-/-} mutant flies so well, which suggests that the mechanism we propose here cannot be essential for centriole duplication.* We understand this confusion and we now make this point more clearly and explain why we think this occurs in more detail (e.g. p22, para.1). We propose that Cdk normally phosphorylates Ana2 to inhibit its ability to promote centriole duplication, but this phosphorylation does not entirely block this function. So, if all other elements of the system are functional, Ana2(12A) is recruited to centrioles for longer than normal, but this does not dramatically perturb centriole duplication because the many other factors that regulate centriole duplication (such as the pulse of Plk4 recruitment to centrioles [Aydogan et al., *Cell*, 2020]) still occur normally and are sufficient to ensure that centrioles still duplicate normally. When Ana2 phosphorylation is mimicked [Ana2(12D/E)], the ability of Ana2 to promote centriole duplication is perturbed (but not abolished). This perturbation is lethal in the early embryo—where the centrioles must duplicate in just a few minutes to keep pace with the rapid nuclear divisions. In somatic cells S-phase is much longer, so these cells can still duplicate their centrioles (as we observe) even though Ana2(12D/E) does not function efficiently. As we now explain, this phenotype (being lethal in the early embryo, but not in somatic cells) is a common feature of mutations that influence the *efficiency* of centriole and centrosome assembly (p17, para.2).

4A. *The reviewer asks us to comment in more detail on why centrioles do not seem to be elongated in the Ana2(12A) mutant wing disc cells (now Figure S8C), even though we show that Ana2(12A) (Figure 4A), and also Sas-6 (Figure 5), are recruited to centrioles for an abnormally long period.* This is an excellent question and, although we do not know the answer, we now discuss this interesting point in more detail (p16, para.1). We think this is likely due to the “homeostatic” nature of centriole growth: in our hands, almost any perturbation that makes centrioles grow for a longer/shorter period, also makes them grow more slowly/quickly, so that they tend to grow to a similar size (Aydogan et al., *JCB*, 2018; *Cell*, 2020). This is fascinating, but poorly understood. When we perturb the system by expressing Ana2(12A), both Ana2(12A) and Sas-6 incorporate into centrioles for a longer period, as we predict (Figure 4A and 5A). Unexpectedly, however, Sas-6 is also recruited to centrioles much more slowly. Thus, as so often

happens, when we perturb the system so the centrioles grow for a longer time, the centrioles “adapt” by growing more slowly. We do not currently understand why this occurs (although we speculate that Ana2 may also be regulated by Cdk/Cyclins to help recruit Sas-6 to centrioles in early S-phase). In the embryo, where S-phase is very short, this homeostatic compensation is not perfect, and the centrioles appear to actually be shorter than normal. In somatic wing-disc cells, where S-phase is much longer, we suspect that there is more scope for homeostatic compensation and so the centrioles grow to the correct size.

4B. In this point (also labelled [4] by the reviewer, so we have retained this numbering but labelled the points A and B) the reviewer asks why levels of Ana2(12A) eventually decline at centrioles once the embryos actually enter mitosis. The reviewer notes our rheostat theory, but suggests a discussion of other mechanisms might be interesting. This is a good point, and we agree that the observation that Ana2(12A) levels ultimately still decline at centrioles during mitosis is likely to be important in explaining why centriole duplication is not more dramatically perturbed by Ana2(12A). We now expand our discussion of this point, highlighting that other mechanisms must help to ensure that Ana2 is not recruited to centrioles during M-phase, and discussing the possibility that the receptors that recruit Ana2 to centrioles are themselves inactivated during mitosis by high levels of Cdk activity (p15, para.1). In such a model, the rapid drop in WT Ana2 centriolar levels is due to a combination of switching off Ana2’s ability to bind to centrioles (as we propose here) *and* switching off the ability of the centrioles to recruit Ana2. For Ana2(12A), only the latter mechanism would operate, so Ana2(12A) levels would start to drop later in the cycle (as the inflexion point at which Ana2 recruitment and loss balances out would be moved to later in the cycle), and these levels would drop more slowly—as we observe.

*5. The reviewer is confused to how the Ana2(12D/E) mutant can rescue the mutant phenotype when it is recruited to centrioles so poorly. Ana2(12D/E) is indeed recruited very poorly to centrioles in the experiment shown in Figure 7. However, this experiment had to be conducted in the presence of WT untagged Ana2—as the embryos do not develop in the presence of only Ana2(12D/E). We would predict that WT Ana2 would bind more efficiently to centrioles than Ana2(12D/E) (which appears to behave as if it has been phosphorylated by Cdk/Cyclins, and so cannot be recruited to centrioles efficiently). Thus, in the experiment we show in Figure 7, the Ana2(12D/E) protein is probably being “outcompeted” for binding to the centriole by the WT protein. In somatic cells expressing *only* Ana2(12D/E) presumably sufficient mutant protein can be recruited to centrioles to support normal centriole duplication (as it no longer has to compete with the WT protein). We now explain our thinking on this point (p18, para.1).*

6. The reviewer wonders whether Ana2(12D/E) may be unable to homo-oligomerize, and this may explain why the protein is not recruited to centrioles efficiently even in the presence of WT protein. This is indeed a possibility, but we think it unlikely as it is widely believed that Ana2/STIL proteins must multimerize to be functional (Arquint et al., *eLife*, 2015; Cottee et al., *eLife*, 2015; Rogala et al., *eLife*, 2015; David et al., *Sci. Rep.*, 2016). As Ana2(12D/E) strongly restores centriole duplication in *ana2^{-/-}* mutant somatic cells, it seems unlikely that it cannot multimerize. Nevertheless, we now specifically highlight that the 12D/E (and 12A) mutations might alter the ability of Ana2 to multimerise (p21, para.2).

Full Revision

Review
COMMONS

We thank the reviewers again for their thoughtful and constructive comments. We hope they will agree that the revised manuscript is now improved and would be appropriate for publication in *The Journal of Cell Biology*.

With best wishes,

June 13, 2022

RE: JCB Manuscript #202205058T

Prof. Jordan W. Raff
University of Oxford
Sir William Dunn School of Pathology
South Parks Road
Oxford OX1 3RE
United Kingdom

Dear Prof. Raff,

Thank you for submitting your revised manuscript entitled "Centriole growth is not limited by a finite pool of components, but is limited by the Cdk1/Cyclin-dependent phosphorylation of Ana2/STIL". We would be happy to publish your paper in JCB pending final revisions necessary to meet our formatting guidelines (see details below). In your final revision, please be sure to address reviewer #2's final minor concerns.

A. MANUSCRIPT ORGANIZATION AND FORMATTING:

- 1) Text limits: Character count for Articles is < 40,000, not including spaces. Count includes abstract, introduction, results, discussion, and acknowledgments. Count does not include title page, figure legends, materials and methods, references, tables, or supplemental legends.
- 2) Figures limits: Articles may have up to 10 main text figures.
- 3) Figure formatting: Scale bars must be present on all microscopy images, including inset magnifications. Molecular weight or nucleic acid size markers must be included on all gel electrophoresis.
- 4) Statistical analysis: Error bars on graphic representations of numerical data must be clearly described in the figure legend. The number of independent data points (n) represented in a graph must be indicated in the legend. Statistical methods should be explained in full in the materials and methods. For figures presenting pooled data the statistical measure should be defined in the figure legends. Please also be sure to indicate the statistical tests used in each of your experiments (either in the figure legend itself or in a separate methods section) as well as the parameters of the test (for example, if you ran a t-test, please indicate if it was one- or two-sided, etc.). Also, if you used parametric tests, please indicate if the data distribution was tested for normality (and if so, how). If not, you must state something to the effect that "Data distribution was assumed to be normal but this was not formally tested."
- 5) * Abstract and title: The abstract should be no longer than 160 words and should communicate the significance of the paper for a general audience. The title should be less than 100 characters including spaces. Make the title concise but accessible to a general readership. *
- 6) Materials and methods: Should be comprehensive and not simply reference a previous publication for details on how an experiment was performed. * Please provide full descriptions in the text for readers who may not have access to referenced manuscripts. *
- 7) Please be sure to provide the sequences for all of your primers/oligos and RNAi constructs in the materials and methods. You must also indicate in the methods the source, species, and catalog numbers (where appropriate) for all of your antibodies. Please also indicate the acquisition and quantification methods for immunoblotting/western blots.
- 8) Microscope image acquisition: The following information must be provided about the acquisition and processing of images:
 - a. Make and model of microscope
 - b. Type, magnification, and numerical aperture of the objective lenses
 - c. Temperature
 - d. Imaging medium
 - e. Fluorochromes

f. Camera make and model

g. Acquisition software

h. Any software used for image processing subsequent to data acquisition. Please include details and types of operations involved (e.g., type of deconvolution, 3D reconstitutions, surface or volume rendering, gamma adjustments, etc.).

10) * Supplemental materials: There are strict limits on the allowable amount of supplemental data. Articles may have up to 5 supplemental figures, therefore please reduce your SI count, for example by combining figures and/or moving some SI data to the main text. Please be sure to correct the callouts in the text accordingly. Please also note that tables, like figures, should be provided as individual, editable files. A summary of all supplemental material should appear at the end of the Materials and methods section. *

13) ORCID IDs: ORCID IDs are unique identifiers allowing researchers to create a record of their various scholarly contributions in a single place. At resubmission of your final files, please consider providing an ORCID ID for as many contributing authors as possible.

Please note that JCB now requires authors to submit Source Data used to generate figures containing gels and Western blots with all revised manuscripts. This Source Data consists of fully uncropped and unprocessed images for each gel/blot displayed in the main and supplemental figures. Since your paper includes cropped gel and/or blot images, please be sure to provide one Source Data file for each figure that contains gels and/or blots along with your revised manuscript files. File names for Source Data figures should be alphanumeric without any spaces or special characters (i.e., SourceDataF#, where F# refers to the associated main figure number or SourceDataFS# for those associated with Supplementary figures). The lanes of the gels/blots should be labeled as they are in the associated figure, the place where cropping was applied should be marked (with a box), and molecular weight/size standards should be labeled wherever possible.

B. FINAL FILES:

****The license to publish form must be signed before your manuscript can be sent to production. A link to the electronic license to publish form will be sent to the corresponding author only. Please take a moment to check your funder requirements before choosing the appropriate license.****

Thank you for this interesting contribution, we look forward to publishing your paper in Journal of Cell Biology.

Sincerely,

Arshad Desai, PhD
Monitoring Editor

Andrea L. Marat, PhD
Senior Scientific Editor

Journal of Cell Biology

Reviewer #1 (Comments to the Authors (Required)):

The authors have convincingly revised their manuscript or explained why they are not able to perform the suggested experiments. Although a number of points raised by this reviewer could not be performed as it was not possible for to generate soluble recombinant Ana2 and Sas-6 proteins, this work is strong and convincing. I am in favor of publishing this study in the JCB.

Reviewer #2 (Comments to the Authors (Required)):

In the revised version of the manuscript, the authors largely responded to the reviewers' concerns with changes to the text. I think the revised manuscript is scientifically sound and can be published in its current form. While the manuscript will be of interest to some in the centrosome field, I feel the overall mechanistic advance is modest.

Minor point

The authors argue that showing the Cdk1 phosphorylation sites are able to be phosphorylated by Cdk1 is not particularly informative. I find this difficult to understand. Some of the putative phosphorylation sites may not be phosphorylated. Instead, the Serine/Threonine could have a structural role that helps protein folding, and mutation of these sites would then not be monitoring the effect of preventing Ana 2 phosphorylation.

University of Oxford, South Parks Road, Oxford OX1 3RE

Professor Jordan Raff
César Milstein Chair of Cancer Biology
Tel: (+44) 01865 275533 (direct)
jordan.raff@path.ox.ac.uk

Melissa Wright
PA to Professor Raff
Tel: (+44) 01865 27559
Fax: (+44) 01865 275515
melissa.wright@path.ox.ac.uk

27th June, 2022

Dear Arshad and Andrea,

Thank you for sending us the reviewer's comments on our revised manuscript. We are delighted that it has been accepted, and we have now reformatted the manuscript to conform with JCB guidelines.

In addition, Reviewer #2 found it hard to understand our argument that determining which potential Cdk1 phosphorylation sites in Ana2 can be phosphorylated by Cdk1/CyclinB in vitro may not be particularly informative. We apologise for not explaining our thinking more clearly. Our main reasoning is that it is widely accepted in the field that the ability/inability of a kinase to phosphorylate a short peptide sequence in vitro does not definitively prove whether that sequence is/is not phosphorylated by that kinase in the context of the whole protein and in the in vivo cellular environment. For this reason, a more extensive analysis using a combination of approaches (including in vivo experiments) will be required to definitively identify the relevant Cdk/Cyclin sites in Ana2. We believe the extensive in vitro analysis suggested by the reviewer cannot, on its own, address this point.

Thank you for all your help with the manuscript.

With best wishes,